# WHEN IS DIVERSITY REWARDED IN COOPERATIVE MULTI-AGENT LEARNING?

**Michael Amir**[*]  **Matteo Bettini**[*]  **Amanda Prorok**
Department of Computer Science and Technology
University of Cambridge
{ma2151,mb2389,asp45}@cl.cam.ac.uk

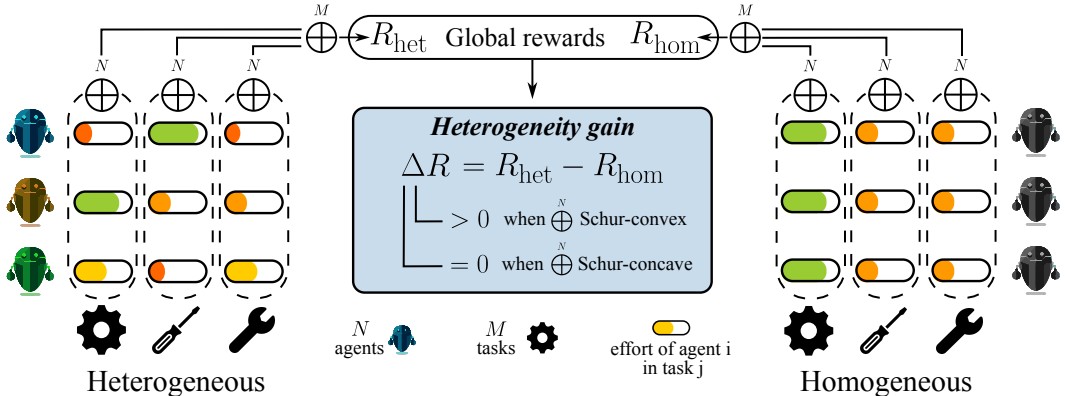

Figure 1: We study and categorize what reward structures lead to the need for behavioral heterogeneity in multi-agent multi-task environments.

## ABSTRACT

The success of teams in robotics, nature, and society often depends on the division of labor among diverse specialists; however, a principled explanation for *when* such diversity surpasses a homogeneous team is still missing. Focusing on multi-agent task allocation problems, we study this question from the perspective of reward design: what kinds of objectives are best suited for heterogeneous teams? We first consider an instantaneous, non-spatial setting where the global reward is built by two generalized aggregation operators: an *inner* operator that maps the $N$ agents' effort allocations on individual tasks to a task score, and an *outer* operator that merges the $M$ task scores into the global team reward. We prove that the curvature of these operators determines whether heterogeneity can increase reward, and that for broad reward families this collapses to a simple convexity test. Next, we ask what incentivizes heterogeneity to *emerge* when embodied, time-extended agents must *learn* an effort allocation policy. To study heterogeneity in such settings, we use multi-agent reinforcement learning (MARL) as our computational paradigm, and introduce *Heterogeneity Gain Parameter Search (HetGPS)*, a gradient-based algorithm that optimizes the parameter space of underspecified MARL environments to find scenarios where heterogeneity is advantageous. Across different environments, we show that HetGPS rediscovers the reward regimes predicted by our theory to maximize the advantage of heterogeneity, both validating HetGPS and connecting our theoretical insights to reward design in MARL. Together, these results help us understand when behavioral diversity delivers a measurable benefit.

---

[*]Equal contribution, listed alphabetically.
Website and code: https://sites.google.com/view/hetgps

# 1 INTRODUCTION

Collective systems, from robot fleets to insect colonies, tend to adopt one of two structures: a uniform shared blueprint or a set of distinct, specialized roles. In multi-agent learning, this is reflected in the choice between behavioral homogeneity (all agents behave identically) and behavioral heterogeneity (agents specialize) (Bettini et al., 2023; 2025; Rudolph et al., 2021). Such behavioral diversity can be achieved, e.g., via distinct policies (neural heterogeneity) or shared policies conditioning on diverse inputs, such as agent roles (Leibo et al., 2019). Although diversity unlocks role specialization and asymmetric information use, it also introduces extra coordination cost, representation overhead, and learning complexity (Li et al., 2021). This trade-off leads us to ask: under what conditions will heterogeneous agents outperform the best homogeneous baseline?

A natural setting to study this question in is *multi-agent task allocation*, where $N$ agents allocate effort across $M$ concurrent tasks. Here, we define *effort* as an *abstract quantity* representing the agent's contribution to a given task (e.g., proximity to a goal, or quantity of a task-specific resource the agent gathered), computed in an environment-specific manner. The focus of our work is *behavioral, outcome-based* heterogeneity, defined through these efforts: a homogeneous team is one where all agents have the same effort allocations (e.g., every agent allocates $0.75$ of its effort to task A and $0.25$ to task B), whereas a heterogeneous team allows agents to achieve specialized allocations. We relate this abstract effort to environmental rewards in many diverse environments, including cooperative navigation, tag, football (Sec. 5), Colonel Blotto games, and level-based foraging (App. E) (Roberson, 2006; Noel, 2022; Papoudakis et al., 2021; Terry et al., 2021). We ask: what effort-based reward functions require heterogeneous behaviors to be maximized?

**Theoretical Insights.** We first study a pure, non-spatial and instantaneous variant of multi-agent task allocation: each agent commits its effort allocation once, and the team is rewarded immediately (Sec. 2). We start from the observation that team reward in many effort–allocation problems can be expressed as $R(A) = U\big(T_1(a_1), \ldots, T_M(a_M)\big)$, where $A = (r_{ij})$ is the $N \times M$ matrix of agent effort allocations, and $a_i$ is the effort allocation vector of agent $i$. The inner, task-level operator $T_i$ assigns a score corresponding to the $N$ agents' efforts on the $i$th task and the outer operator $U$ combines the resulting $M$ task scores into a scalar global reward. Choosing $T$ and $U$ to be the sum operator $\sum$ recovers the $\sum_j \sum_i r_{ij}$ reward common in RL, whereas alternatives such as MAX, MIN, power means, or soft-max encode very different effort–reward relationships. Assuming such a reward structure, we compare the optimal heterogeneous reward, $R_{\text{het}}$, with the best reward attainable under a homogeneous allocation, $R_{\text{hom}}$, and define their difference as the *heterogeneity gain* $\Delta R = R_{\text{het}} - R_{\text{hom}}$ (Fig. 1). Our main insight is that $\Delta R$ is determined by the *curvature* of $T$ and $U$: specifically, whether they are *Schur-convex* or *Schur-concave*. These criteria immediately enable us to characterize the heterogeneity gain of broad families of reward functions (Table 3); for instance, the soft-max operator switches from Schur-concave to Schur-convex as its temperature increases. We also find exact expressions for $\Delta R$ in several important cases. These results help explain, for example, why a reward structure that involves a min operator (usually used to enforce that only one agent should pursue a goal) will require behavioral diversity from the agents (Bettini et al., 2024). We relate our findings to multi-agent reinforcement learning (MARL), where environments may be embodied and time-extended, by setting $R(A_t)$ as the stepwise reward over an allocation sequence $(A_t)_{t=1,\ldots,T}$.

**Algorithmic Search.** To study heterogeneity in MARL settings not covered by our theoretical analysis, we develop *Heterogeneity Gain Parameter Search* (HetGPS), a gradient-based algorithm that optimizes parameters $\theta$ of underspecified, differentiable MARL environments via backpropagation to find configurations that maximize or minimize the empirical $\Delta R$ (we assume differentiability for training efficiency, but consider non-differentiable environments in App. P). While HetGPS can in principle optimize any differentiable environment feature to influence $\Delta R$, we use it here to explore reward structures, as a means of verifying and extending our theoretical insights. Maximizing the heterogeneity gain allows us to discover reward functions where behavioral diversity is essential. Minimizing the gain leads us to settings where homogeneous policies are sufficient.

**Experiments.** We validate our theoretical insights, and HetGPS, in simulation, by evaluating in both single-shot and long-horizon reinforcement learning environments whose reward structure instantiates the kinds of aggregation operators studied. First, in a continuous and a discrete matrix game, we test reward structures based on all nine possible combinations of $\{\min, \text{mean}, \max\}$, and find that the heterogeneity gains that result from the agents' learned policies match our theoretical predictions: concave outer operators and convex inner operators benefit heterogeneous teams. Next, we test

the same operators in embodied, partially observable environments: Multi-goal-capture, Tag, and Football. We find that our theory also transfers to such long-horizon MARL settings, and show that reward structures that maximize heterogeneity are meaningful and practically useful. Finally, we find that the empirical heterogeneity gain disappears as the richness of agents' observations is increased, recovering the finding that rich observations allow agents with identical policy networks to be behaviorally heterogeneous (Bettini et al., 2023; Leibo et al., 2019).

We then turn to HetGPS. Across two parameterizable families of operators (Softmax and Power-Sum), we show that, despite running on embodied environments, HetGPS rediscovers the reward regimes predicted by our curvature theory to maximize the heterogeneity gain, validating both HetGPS and the connection between our theoretical insights and MARL reward design (Sec. 5).

## 1.1 RELATED WORKS

**Behavioral Diversity in MARL.** Behavioral heterogeneity, where capability-identical agents learn distinct policies, can markedly improve exploration, robustness, and reward (Bettini et al., 2023). Yet heterogeneity reduces parameter sharing and thus sample-efficiency, so a core practical question is *when* its benefits outweigh that cost. Existing MARL methods typically adopt one of two poles: endowing each agent with its own network, or enforcing parameter sharing so all agents follow a single policy (Gupta et al., 2017a; Rashid et al., 2020; Foerster et al., 2018; Kortvelesy & Prorok, 2022; Sukhbaatar et al., 2016). A large body of work explores the efficiency–diversity trade-off (Christianos et al., 2021; Fu et al., 2022) by interpolating between these extremes: e.g., injecting agent IDs into the observation (Foerster et al., 2016; Gupta et al., 2017a), masking different subsets of shared weights (Li et al., 2024b), sharing only selected layers (Li et al., 2021), pruning a shared network into agent-specific sub-graphs (Kim & Sung, 2023), or producing per-agent parameters with a hypernetwork (Tessera et al., 2024). Further, several methods for promoting behavioral diversity in MARL have been proposed, such as: conditioning agents' policies on a latent representation (Wang et al., 2020), decomposing and clustering action spaces (Wang et al., 2021), dynamically grouping agents to share parameters (Yang et al., 2022), applying structural constraints to the agents' policies (Bettini et al., 2024), or by intrinsic rewards that maximize diversity (Li et al., 2021; Jaques et al., 2019; Wang et al., 2019; Jiang & Lu, 2021; Mahajan et al., 2019; Liu et al., 2023; 2024; Li et al., 2024a). While these studies demonstrate *how* to obtain diversity, they presume tasks where heterogeneity is advantageous. Our work addresses the orthogonal question of *when* diversity is beneficial, giving a principled characterization of which reward structures create that incentive in the first place.

**Task Allocation.** Classic resource–allocation settings, in which a team must divide finite effort among simultaneous objectives, are a central proving ground for cooperative MARL. In robotics, potential-field and market-based learning are the dominant tools for coverage, exploration, and load-balancing tasks (Gupta et al., 2017b; Lowe et al., 2017). Game-theoretic analysis and, recently, MARL, play the same role in discrete counterparts such as Colonel-Blotto contests, where players decide how to spread forces over several "battlefields" (Roberson, 2006; Noel, 2022). Embodied benchmarks like level-based foraging are heavily studied in MARL, and expose the tension between uniform and specialized effort allocations (Papoudakis et al., 2021). The survey of (Zhang et al., 2019) highlights how cooperative performance is governed by the shape of the shared reward and the equilibria it induces. Our contribution sharpens this perspective: we prove that the *curvature* of nested aggregation operators characterizes when heterogeneous allocations dominate homogeneous ones, and introduce algorithmic tools for further exploring settings where diversity is needed.

**Environment Co-design.** Co-design is a paradigm where agent policies *and* their mission or environment are simultaneously optimized (Gao et al., 2024; Amir et al., 2025). Our HetGPS algorithm is related to PAIRED (Dennis et al., 2020), a method which automatically designs environments in a curriculum such that an *antagonist* agent succeeds while the *protagonist* agent fails. This makes it so that resulting environments are challenging enough without being unsolvable. Similarly, HetGPS designs environments that are advantageous to heterogeneous teams, while disadvantaging homogeneous teams. The key differences are: (1) the environment designer uses direct regret gradient backpropagation via a differentiable simulator instead of RL; this enables higher efficiency by directly leveraging all the environment gradient data available during collection while preventing RL-related issues identified in subsequent works (Jiang et al., 2021; Parker-Holder et al., 2021) such as exploration inefficiency and the need for a reward signal; and (2), the protagonist and antagonist are independent multi-agent teams instead of single agents.

## 2 PROBLEM SETTING

Consider a set of $N$ agents and $M$ tasks. Each agent $i \in \{1, \ldots, N\}$ allocates *effort* among the tasks according to the budget constraints: $r_{i1}, r_{i2}, \ldots, r_{iM} \geq 0$ with $\sum_{j=1}^{M} r_{ij} \leq 1$, where $r_{ij}$ is *defined* as the effort agent $i$ puts into task $j$. Here, "effort" $r_{ij}$ is a scalar input to the reward function representing the agent's contribution to the task, such as resource allocation (App. E) or realized goal proximity (Sec. 5). We can consider both continuous allocations ($r_{ij}$ can be any real number) and discrete allocations ($r_{ij}$ restricted to some finite set of options), with most results in this work focusing on the continuous case. We collect all agents' allocations into an $N \times M$ matrix: $A = [r_{ij}]$[1].

For each task $j$ let the $j$-th column of the effort matrix be $a_j = [r_{1j}, \ldots, r_{Nj}]^\top$. A *task-level aggregator* $T_j : \mathbb{R}^N \to \mathbb{R}$ maps these efforts to a *task score*, and an *outer aggregator* $U : \mathbb{R}^M \to \mathbb{R}$ combines the $M$ scores into the team reward, $R(A) = U\big(T_1(a_1), \ldots, T_M(a_M)\big)$. Both $T_j$ and $U$ are *generalised aggregators*: symmetric and coordinate-wise non-decreasing, mirroring the familiar properties of $\sum$. When every task shares the same inner aggregator we simply drop the subscript and write $T$. In Figure 1, to highlight fact that $R$ is aggregating rewards, we write $R(A) = \bigoplus_{j=1}^{M} \bigoplus_{i=1}^{N} r_{ij}$, where (in abuse of notation) the outer symbol $\bigoplus$ denotes $U$ and the inner symbol $\bigoplus$ denotes $T_j$.

**Homogeneous vs. Heterogeneous Strategies.** A *homogeneous strategy* is one where all agents have the same allocation (i.e., devote the same amount of effort to a given task $j$): $r_{ij} = c_j \forall i, j$. In this case, the allocation matrix $A$ consists of identical rows. We define $R_{\text{hom}} = \max_{(c_1, \ldots, c_M) \in \Delta_{\leq}^{M-1}} R\big(A\big)$ where $\Delta_{\leq}^{M-1} = \{(c_1, \ldots, c_M) \mid c_j \geq 0, \ \sum_j c_j \leq 1\}$ is the closed unit simplex. A *heterogeneous strategy* allows each agent $i$ to choose any $(r_{i1}, \ldots, r_{iM}) \in \Delta_{\leq}^{M-1}$ independently. Then $R_{\text{het}} = \max_{A \in (\Delta_{\leq}^{M-1})^N} R\big(A\big)$. We define the *heterogeneity gain* as: $\Delta R = R_{\text{het}} - R_{\text{hom}}$. This quantity measures how much greater the overall reward can be when agents are allowed to specialize differently across tasks, compared to when they must behave identically. Characterizing when $\Delta R > 0$ is our main focus in this work.

**MARL extension.** In MARL, the effort value $r_{ij}$ represents the contribution of agent $i$ to task $j$ *as computed by the environment based on agent $i$'s actions*. The aggregate reward $R(A)$ can represent: (i) the payoff of a one-shot effort-allocation game, (ii) the return or sparse terminal reward of an episode, or (iii) the stepwise reward, giving the discounted return $\sum_{t=0}^{T} \gamma^t R\big(A_t\big)$ for a sequence $\big(A_t\big)_{t=1, \ldots T}$ of allocations[2]. $\Delta R > 0$ implies that the best heterogeneous policies outperform the best homogeneous ones. In practice, this is *evidence of* an advantage to heterogeneity and not a formal guarantee, as learning agents may not always converge to optimal policies.

**Examples.** App. I contains examples of generalized aggregators. Our framework is flexible, and can be applied to many settings, including ones not ordinarily thought of as "task allocation": in Sec. 5, we apply it to one-shot allocation games, multi-agent navigation, tag, and football. Furthermore, in App. E, we analyze the heterogeneity gain of two well-known environments from the literature: Team Colonel Blotto games (Noel, 2022) and level-based foraging (Papoudakis et al., 2021).

## 3 ANALYSIS

Focusing on continuous allocations, we ask what properties of aggregators guarantee $\Delta R > 0$. We draw on the concept of Schur-convexity. Schur-convex functions can be understood as generalizing symmetric, convex aggregators: every convex and symmetric function is Schur-convex, but a Schur-convex function is not necessarily convex (Roberts & Varberg, 1974; Peajcariaac & Tong, 1992). Proofs for all results are available in App. G.

---

[1] All results in this work can be extended to the case where $r_{i1}, r_{i2}, \ldots, r_{iM} \geq B_{min}$ and $\sum_{j=1}^{M} r_{ij} \leq B_{max}$ for some arbitrary $B_{min}, B_{max} \in \mathbb{R}$.

[2] To extend this further, our theoretical results hold even if the reward function varies over time, $R_t(A_t)$.

Since both the outer aggregator $U$ and the task-level aggregators $T_j$ are non-decreasing, an optimal effort allocation will always have each agents' efforts summing to 1. Hence, from here on, we **assume without loss of generality** that $\sum_{j=1}^{M} r_{ij} = 1$. We call such allocations **admissible**.

**Definition 3.1** (Majorization). *Let $x = (x_1, \ldots, x_N)$ and $y = (y_1, \ldots, y_N)$ be two vectors in $\mathbb{R}^N$ such that $\sum_{i=1}^{N} x_{(i)} = \sum_{i=1}^{N} y_{(i)}$. Let $x_{(1)} \geq x_{(2)} \geq \cdots \geq x_{(N)}$ and $y_{(1)} \geq y_{(2)} \geq \cdots \geq y_{(N)}$ be the components of $x$ and $y$ sorted in descending order. We say that $x$* majorizes *$y$ (written $x \succ y$) if $\sum_{i=1}^{k} x_{(i)} \geq \sum_{i=1}^{k} y_{(i)}$ for $k = 1, 2, \ldots, N - 1, N$.*

**Definition 3.2** (Schur-Convex Function). *A symmetric function $f : \mathbb{R}^N \to \mathbb{R}$ is* Schur-convex *if for any two vectors $x, y \in \mathbb{R}^N$ with $x \succ y$, we have $f(x) \geq f(y)$. If the inequality is strict whenever $x$ and $y$ are not permutations of each other, then $f$ is said to be* strictly Schur-convex. *Similarly, $f$ is* Schur-concave *if $f(x) \leq f(y)$ whenever $x \succ y$.*

Intuitively, $x \succ y$ means one can obtain $y$ from $x$ by repeatedly moving mass from larger to smaller coordinates, thereby making the vector more uniform. Schur-convexity is then a statement on a function's *curvature*: $f$ is *Schur-convex* if it increases with inequality, or is *Schur-concave* if it increases with uniformity. We show here a connection between Schur-convexity (concavity) and $\Delta R$.

Call an allocation matrix $A$ *trivial* if there exists a task $j^\star$ such that every agent allocates its entire budget to that task, i.e. $r_{ij^\star} = B_{\max}$ and $r_{ij} = 0 \ \forall i, \ \forall j \neq j^\star$; otherwise $A$ is *non-trivial*. Then:

**Theorem 3.1** (Positive Heterogeneity Gain via Schur-convex Inner Aggregators). *Let $N, M \geq 2$, and assume that (i) each* task-level aggregator $T_j$ *is strictly Schur-convex and (ii) the* outer aggregator *$U$ is coordinate-wise strictly increasing. Then either all admissible optimal homogeneous allocations are trivial, or $\Delta R > 0$.*

If the task-level aggregator is instead Schur-concave, we can show there is no heterogeneity gain:

**Theorem 3.2** (No Heterogeneity Gain via Schur-concave Inner Aggregators). *Let $N, M \geq 2$. If each task-level aggregator $T_j$ is* Schur-concave *then $\Delta R = 0$.*

We see that Schur-convexity of the inner aggregator produces $\Delta R > 0$, whereas Schur-concavity implies $\Delta R = 0$. Analyzing the outer aggregator $U$ is trickier, because it acts on task-score vectors $\left(T_1(a_1), \ldots, T_M(a_M)\right)$ whose sum $\sum_{i=1}^{M} T_i(a_i)$ may vary, so majorization is not directly applicable. However, we can extend our analysis to $U$ if our inner aggregators are *normalized* to keep the sum constant: $\sum_{i=1}^{M} T_i(a_i) = C$ for any admissible allocation. Assuming this, we can invoke majorization again, and the relationship between convexity and $\Delta R$ reverses: if the outer aggregator $U$ is Schur-convex, the heterogeneity gain vanishes. Let us prove this.

**Theorem 3.3** (No Heterogeneity Gain for Schur-Convex $U$ with Constant-Sum Task Scores). *Let $N, M \geq 2$. Suppose that for any admissible allocation $A$, (i) every task score is non-negative, and obeys $T_i(0, \ldots, 0) = 0$, and (ii) the sum of task score is always $\sum_{j=1}^{M} T_j\left(a_j\right) = C$. If $U$ is strictly Schur-convex function, then $\Delta R = 0$.*

It is crucial to note that this constant-sum assumption is specific to Thm. 3.3 and is not required for our other results, which apply broadly.

**Sum-Form Aggregators.** In App. F, we show that the above results reduce to a simple convexity test for *sum-form aggregators*: a broad class of aggregators that describes most reward structures we consider in this work. This makes testing whether $\Delta R > 0$ a simple computation in many cases.

**Parameterizable Families of Aggregators.** A core topic of this work is *reward design*: how can we craft team objectives that either advantage or disadvantage behavioral diversity? To do this, it is helpful to first identify an appropriate search space. Our theoretical analysis enables us to narrow down this search space, and focus on aggregators whose *curvature* can be parametrized. Many *family of aggregator functions* $\{f_t(\cdot)\}_{t \in \mathbb{R}}$ can be parametrized by a scalar $t$ which controls whether the aggregation is *Schur-convex* or *Schur-concave*, and how strongly it penalizes (or favors) inequalities among the components. For example, the *softmax aggregator* $\sum_{i=1}^{N} \frac{\exp\left(t \cdot r_{ij}\right)}{\sum_{\ell=1}^{N} \exp\left(t \cdot r_{\ell j}\right)}$ is parametrized by its temperature, $t$, transitioning from being strictly Schur-concave when $t < 0$ to strictly Schur-convex when $t > 0$. We can define a space of reward functions by selecting both the

Discrete and continuous heterogeneity gains

|  | $T = \min$ | $T = \text{mean}$ | $T = \max$ |
|---|---|---|---|
| | | *Outer* $U = \min$ | |
| $\Delta R_{\mathrm{F}}$ | 0 | 0 | $(M-1)/M$ |
| $\Delta R_{\mathrm{D}}$ | 0 | $\lfloor N/M \rfloor/N$ | $\mathbb{1}_{\{N \geq M\}}$ |
| | | *Outer* $U = \text{mean}$ | |
| $\Delta R_{\mathrm{F}}$ | 0 | 0 | $(M-1)/M$ |
| $\Delta R_{\mathrm{D}}$ | 0 | 0 | $(\min\{M,N\}-1)/M$ |
| | | *Outer* $U = \max$ | |
| $\Delta R_{\mathrm{F}}$ | 0 | 0 | 0 |
| $\Delta R_{\mathrm{D}}$ | 0 | 0 | 0 |

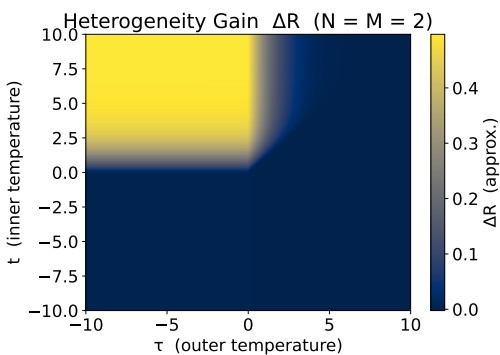

Figure 2: **Left:** Discrete ($\Delta R_{\mathrm{D}}$) and continuous-allocation ($\Delta R_{\mathrm{F}}$) heterogeneity gains for all $U, T \in \{\min, \text{mean}, \max\}$. The indicator $\mathbb{1}_{\{N \geq M\}}$ equals 1 if $N \geq M$ and 0 otherwise. **Right:** We plot the parametrized heterogeneity gains $\Delta R(t, \tau; N)$ when $U$ and $T$ are soft-max aggregators.

task scores and outer aggregator to be softmax functions: let $T_j(A) = \sum_{i=1}^{N} \frac{\exp(t \cdot r_{ij})}{\sum_{\ell=1}^{N} \exp(t \cdot r_{\ell j})} \, r_{ij}$,

and let $U(T_1(a_1), \ldots T_M(a_m)) = \sum_{j=1}^{M} \frac{\exp(\tau \cdot T_j(A))}{\sum_{\ell=1}^{M} \exp(\tau \cdot T_\ell(A))} \, T_j(A)$, where $t, \tau \in \mathbb{R}$ parametrize the inner and outer aggregators, respectively. $\Delta R$ is then dependent on $t$ and $\tau$. Fig. 2 plots $\Delta R$ when $N = M = 2$. As a case study, we derive lower bounds on $\Delta R$ when $N = M$ in Thm. 3.4.

**Theorem 3.4** (Softmax heterogeneity gain for $N = M$). *Assume $N = M \geq 2$, and let $\sigma(t, N) := \frac{e^t}{e^t + N - 1}$. The heterogeneity gain for softmax aggregators **(i)** equals $\Delta R(t, \tau; N) = 0$ when $t \leq 0$; **(ii)** is bounded below by $\sigma(t, N) - \frac{1}{N}$ when $t > 0, \tau \leq 0$; and **(iii)** is bounded below by $\max\{\sigma(t, N) - \sigma(\tau, N), 0\}$ when $t > 0, \tau \geq 0$.*

Tab. 3 contains more examples of aggregation operators parameterized by $t$. These families provide a search space for potential reward functions, allowing us to sweep smoothly from $\Delta R = 0$ to $\Delta R > 0$ reward regimes. As $t \to \pm\infty$, most such aggregators converge to either min or max, and often reduce to the arithmetic mean for certain parameter choices, motivating us to ask what the heterogeneity gain is when the outer and inner aggregator belong to the set $\{\min, \text{mean}, \max\}$. These aggregators are of special interest, since "min" can be seen as a "maximally" Schur-concave function, "max" can be seen as a "maximally" Schur-convex function, and "mean" is both Schur-convex and Schur-concave. Hence, it is worth asking what the heterogeneity gain is when the outer and inner aggregator belong to the set $\{\min, \text{mean}, \max\}$. We derive these gains in two cases: continuous allocations where $r_{ij} \in [0, 1]$, and discrete effort allocations where $r_{ij} \in \{0, 1\}$. The results are summarized in Fig. 2, lefthand side (formal derivation available in App. G).

## 4 HETEROGENEITY GAIN PARAMETER SEARCH (HETGPS)

In complex scenarios where theory might be less applicable, we study heterogeneity through algorithmic search. We consider the setting of a Parametrized Dec-POMDP (PDec-POMDP, defined in App. O). A PDec-POMDP represents a Dec-POMDP (Oliehoek et al., 2016), where the observations, transitions, or reward are conditioned on parameters $\theta$. Hence, the return obtained by the agents, $G^\theta(\pi)$, can be differentiated with respect to $\theta$: $\nabla_\theta G^\theta(\pi) = \frac{\partial}{\partial \theta} G^\theta(\pi)$. In particular, computing this gradient in a differentiable simulator allows us to back-propagate through time and optimize $\theta$ via gradient ascent[3].

**Heterogeneity Gain Parameter Search (HetGPS).** We now consider the problem of learning the environment parameters $\theta$ to maximize the *empirical* heterogeneity gain. The empirical heterogeneity gain is defined as the difference in performance between heterogeneous and homogeneous teams in

---

[3] Although the same approach can train policies (Xu et al., 2022; Song et al., 2024), HetGPS instead optimizes environment parameters and policies separately, using standard zeroth-order policy-gradient methods, to avoid being trapped in local minima.

---

**Algorithm 1** Heterogeneity Gain Parameter Search (HetGPS)

---

**input** Environment parameters $\theta$, environment learning rate $\alpha$, heterogeneous agent policy $\pi_{\text{het}}$, homogeneous agent policy $\pi_{\text{hom}}$
1: **for** $i$ in iterations **do**
2:     $\text{Batch}_{\text{het}}^{\theta} = \text{Rollout}(\theta, \pi_{\text{het}})$ {rollout het policies in environment $\theta$}
3:     $\text{Batch}_{\text{hom}}^{\theta} = \text{Rollout}(\theta, \pi_{\text{hom}})$ {rollout hom policies in environment $\theta$}
4:     $\text{HetGain}^{\theta} = \text{ComputeGain}(\text{Batch}_{\text{het}}^{\theta}, \text{Batch}_{\text{hom}}^{\theta})$
5:     **if** train_env $(i)$ **then**
6:         $\theta \leftarrow \theta + \alpha \nabla_{\theta} \text{HetGain}^{\theta}$ {train environment via backpropagation}
7:     **if** train_agents$(i)$ **then**
8:         $\pi_{\text{het}} \leftarrow \text{MarlTrain}(\pi_{\text{het}}, \text{Batch}_{\text{het}}^{\theta})$ {train het policies via MARL}
9:         $\pi_{\text{hom}} \leftarrow \text{MarlTrain}(\pi_{\text{hom}}, \text{Batch}_{\text{hom}}^{\theta})$ {train hom policies via MARL}
**output** final environment configuration $\theta$, policies $\pi_{\text{het}}, \pi_{\text{hom}}$

---

a given PDec-POMDP parametrization. We compare *neurally heterogeneous* agents (independent parameters) with *neurally homogeneous* agents (shared parameters). We denote their policies as $\pi_{\text{het}}$ and $\pi_{\text{hom}}$. Then, we can simply write the gain as: $\text{HetGain}^{\theta}(\pi_{\text{het}}, \pi_{\text{hom}}) = G^{\theta}(\pi_{\text{het}}) - G^{\theta}(\pi_{\text{hom}})$, representing the return of heterogeneous agents minus that of homogeneous agents on environment parametrization $\theta$. HetGPS, shown in Alg. 1, learns $\theta$ by performing gradient ascent to maximize the gain: $\theta \leftarrow \theta + \alpha \nabla_{\theta} \text{HetGain}^{\theta}(\pi_{\text{het}}, \pi_{\text{hom}})$. The environment and the agents are trained in an iterative, bilevel optimization process. We discuss this process, and *alternatives when the simulator is non-differentiable*, in App. P. At every training iteration, HetGPS collects roll-out batches in the current environment $\theta$ for both heterogeneous and homogeneous teams, computing the heterogeneity gain on the collected data. Then, it updates $\theta$ to maximize the heterogeneity gain. Finally, to train the agents, it uses MARL, with any on-policy algorithm (e.g., MAPPO (Yu et al., 2022)). The functions `train_env` and `train_agents` determine when to train each of the components in HetGPS. We consider two possible training regimes: (1) *alternated*: where HetGPS performs cycles of $x$ agent training iterations followed by $y$ environment training iterations and (2) *concurrent*: where agents train at every iteration and the environment is updated every $x$ iterations. Note that by performing descent instead of ascent, HetGPS can also be used to *minimize* the heterogeneity gain.

## 5 EXPERIMENTS

To empirically ground our theoretical analysis, we conduct a three-stage experimental study in cooperative MARL. We first analyze a one-step, observation-free matrix game in which each agent allocates effort $r_{ij}$ over $M$ tasks, and consider reward structures defined by aggregator pairs $U, T \in \{\min, \text{mean}, \max\}$. We find that the agents' learned policies recover the exact heterogeneity gains derived in the theory (Fig. 2). Next, we transfer the same reward structures into embodied, time-extended environments: Multi-goal-capture, 2v2 Tag, and Football. We show that our curvature theory continues to be informative in these settings. We discuss the learning dynamics that result, and perform further experiments highlighting the difference between *neural* and *behavioral* heterogeneity (Bettini et al., 2023), important for understanding our insights. Finally, to study HetGPS, we parametrize the reward structure of Multi-goal-capture using either parametrized Softmax or Power-Sum aggregators (App. I), and run HetGPS to learn parameterizations that maximize the heterogeneity gain. HetGPS learns the theoretically optimal aggregator instantiations, validating its effectiveness at discovering heterogeneous missions. Implementation details and visualizations are available in App. C and App. K.

**(i) Task Allocation.** We consider a one-step observationless matrix game where $N$ agents need to choose between $M$ tasks. Their actions are effort allocations $r_{ij}$ with $r_{ij} \geq 0, \sum_j r_{ij} = 1$, composing matrix $A$. With aggregators taken from the set $U, T \in \{\min, \text{mean}, \max\}$, our goal is to empirically confirm the heterogeneity gains derived in the theory *in a learning context*. Each time the game is played, all agents obtain the global reward $R(A)$ computed through the double aggregator. We consider two setups: (1) *Continuous* ($r_{ij} \in \mathbb{R}_{0 \leqslant x \leqslant 1}$): agents can distribute their efforts across tasks, (2) *Discrete* ($r_{ij} \in \{0, 1\}$): agents choose only one task. We train with $N = M = 4$ for 12 million steps. Fig. 6 shows the evolution of the heterogeneity gains. The final results match *exactly* the

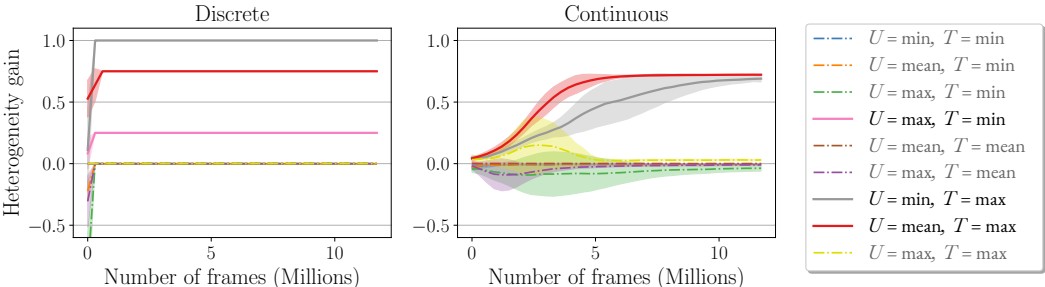

Figure 3: Heterogeneity gain for the discrete and continuous matrix games with $N = M = 4$ over training iterations. We report mean and standard deviation after 12M frames over 9 random seeds. The final results match the theoretical predictions in the Table of Fig. 2. Solid lines indicate reward structures predicted by theory to have $\Delta R > 0$ in either the discrete or continuous setting; dashed lines indicate predicted no gain in both settings. Final gain values are reported in Tab. 7 and Tab. 8.

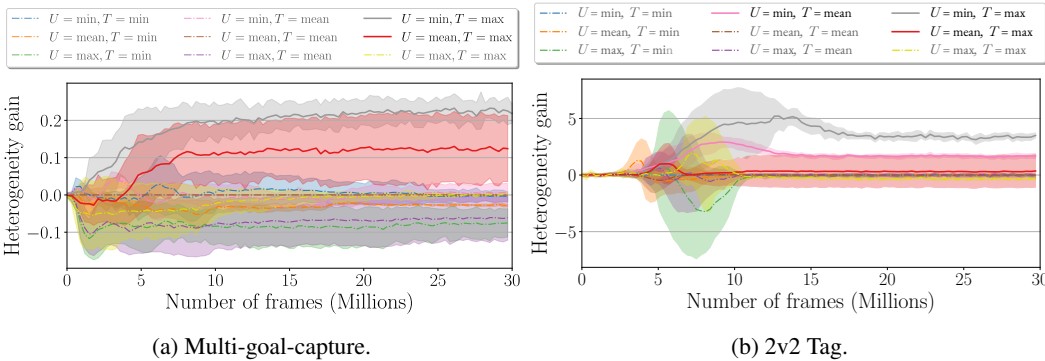

(a) Multi-goal-capture.                    (b) 2v2 Tag.

Figure 4: Heterogeneity gain for Multi-goal-capture and 2v2 Tag throughout training. We report mean and standard deviation for 30 million training frames over 9 random seeds. Final gain values are reported in Tab. 11 and Tab. 12.

theoretical predictions of Fig. 2 and our curvature theory: *concave* outer and *convex* inner aggregators favor heterogeneity. Additional details and results, e.g., for $N, M \in \{2, 8, 11\}$, are in App. J.

**(ii-1) Multi-goal-capture.** Next, we investigate a time-extended, embodied scenario called Multi-goal-capture, based on multi-goal navigation missions (Terry et al., 2021). In Multi-goal-capture, agents need to navigate to goals, and efforts $r_{ij}^t$ are continuous scalars computed based on their proximity to these goals. We provide details in App. K. Our goal is to show that the results obtained in the matrix game still hold in this embodied, long-horizong setting. We again consider aggregators $U, T \in \{\min, \text{mean}, \max\}$. After 30M training frames (Fig. 4a) the empirical heterogeneity gains differ, numerically, from those of the static matrix-game because agents now realize their allocations $r_{ij}$ through time-extended motion. *Nonetheless, our curvature theory reliably predicts when there is a heterogeneity gain* (Fig. 2): it is positive *only* for the concave–convex pairs $U = \min, T = \max$ and $U = \text{mean}, T = \max$. We further explain these results (including the interesting presence of "negative" heterogeneity gains) in App. K. Note that the aggregator pairs in this experiment are not contrived: they encode practically meaningful global objectives. For example, $U = \max, T = \max$ implies "at least one agent should go to at least one goal"; $U = \max, T = \min$ implies "all agents should go to the same goal", and so on. $U = \min, T = \max$, a concave-convex setting shown by our theory to favor heterogeneity, implies "each agent should go to a different goal and all goals should be covered" which is a natural goal for this scenario. This is because $T = \max$ encodes a task that needs just one agent to be completed (e.g., find an object), while $U = \min$ encodes that all tasks should be attended (i.e., agents need to diversify their choices).

*Observability-Heterogeneity Trade-Off:* To understand our theoretical results, it is important to solidify the difference between *neural heterogeneity* (agents having different neural networks) and *behavioral heterogeneity* (agents acting differently). Our insights concern behavioral heterogeneity, which need not be neurally induced. We show this in App. N, showing that: as the observability of

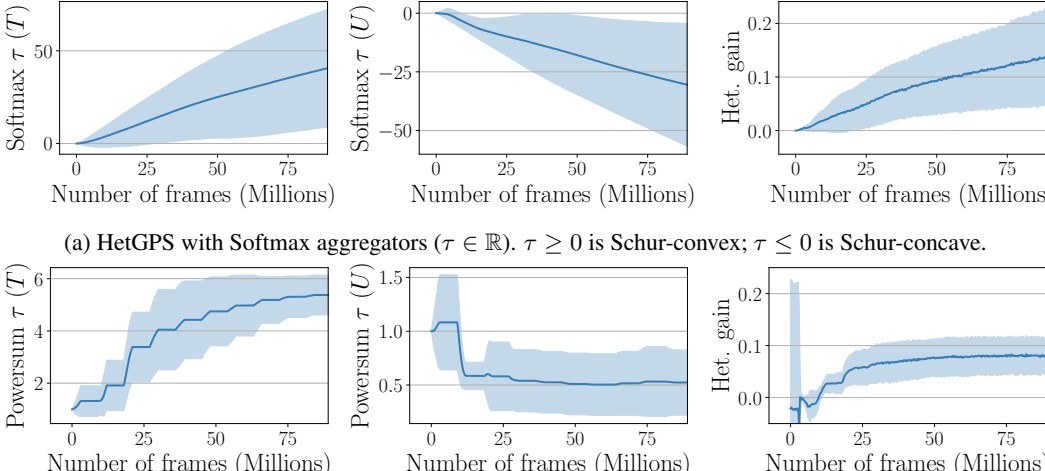

(a) HetGPS with Softmax aggregators ($\tau \in \mathbb{R}$). $\tau \geq 0$ is Schur-convex; $\tau \leq 0$ is Schur-concave.

(b) HetGPS with Power-Sum aggregators ($\tau \in [0.3, 6]$). $\tau \geq 1$ is Schur-convex; $\tau \leq 1$ is Schur-concave.

Figure 5: HetGPS results in Multi-goal-capture. The two leftmost columns report the evolution of aggregator parameters through training, while the rightmost column shows the obtained heterogeneity gain. This result empirically demonstrates that HetGPS rediscovers the reward structure predicted by our theory to maximize the gain, *making the inner aggregator convex, and the outer aggregator concave*. We report mean and standard deviation for 90M training frames over 13 random seeds.

neurally homogeneous agents increases (allowing them to sense each other), these agents can become behaviorally heterogeneous, and thus optimize the heterogeneity gain. This result is visualized here.

**(ii-2) 2v2 Tag.** In our tag experiment, two learning chasers pursue two heuristic escapees in a randomized obstacle field. We define the effort $r_{ij}^t$ to be 1 if chasing agent $i$ manages to capture escaping agent $j$ by time $t$, and 0 otherwise. Whereas Multi-goal-capture had continuous effort allocations, here they are *discrete*. The global reward is again computed with aggregators $U, T \in \{\min, \text{mean}, \max\}$, and is awarded at every time step. This is a *sparse* reward signal only awarded upon mission success. For example, $U = \min, T = \max$ pays out only if both escapees are each caught by a chaser, encouraging heterogeneity. Training outcomes are summarized in Fig. 4b. We again see that our theoretical results in Fig. 2 (discrete efforts) predict *exactly* which aggregators will exhibit $\Delta R > 0$. More details, visuals, and experiments with greater number of agents are available in App. L and here.

**(ii-3) Football.** We evaluate our theory in a complex continuous control football game to explore what happens when our reward structure $R(A)$ is only part of a global cooperative reward. To this end, we design a drill in the VMAS Football environment (Bettini et al., 2022), where one agent is tasked to score, while the other has to block the incoming opponent. App. M shows that, also in this case, our theory is highly predictive, with visuals available here.

**(iii) Heterogeneous Reward Design.** We apply HetGPS to Multi-goal-capture, and ask whether it finds the same aggregator parameterizations predicted by our theory to maximize the heterogeneity gain. We turn the environment into a PDec-POMDP by parameterizing the reward as $R^\theta(A^t) = \bigoplus_{j=1}^{M}{}^\theta \bigoplus_{i=1}^{N}{}^\theta r_{ij}^t$, with parametrized inner and outer aggregators $U^\theta = \bigoplus^\theta, T^\theta = \bigoplus^\theta$. Our goal is to learn the parameters $\theta = (\tau_1, \tau_2)$, parametrizing $T$ and $U$ respectively, that maximize the heterogeneity gain. We consider two parametrized aggregators from Tab. 3: Softmax and Power-Sum. *Softmax*: we parameterize both $U^\theta$ and $T^\theta$ using Softmax. We initialize $\tau_1 = \tau_2 = 0$, so $U$ and $T$ are initially *mean*, and run HetGPS (in App. Q we show that HetGPS is robust to adversarial initializations). In Fig. 5a, we show that, to maximize the heterogeneity gain, HetGPS learns to maximize $\tau_1$, making $T$ Schur-convex, while minimizing $\tau_2$, making $U$ Schur-concave. Hence, it rediscovers the theoretically optimal reward function. The large variance in final parameters occurs because the Softmax aggregator saturates for large magnitudes (e.g., $|\tau| > 5$); HetGPS *correctly* identifies this, leading seeds to converge to arbitrary large values within it. *Power-Sum*: we parametrize both aggregators with Power-Sum. We initialize both functions to $\tau_1 = \tau_2 = 1$, representing *sum*, and run HetGPS. We constrain $\tau_{1,2} \in [0.3, 6]$ to stabilize learning. In Fig. 5b, we

show that HetGPS learns to maximize $\tau_1$, making $T$ Schur-convex, while minimizing $\tau_2$, making $U$ Schur-concave; again rediscovering the optimal parametrization our theory predicts. These results simultaneously validate HetGPS and our curvature theory, since each arrives at the same reward structure independently.

## 6 DISCUSSION

This work introduces tools for both *diagnosing* and *designing* reward functions that incentivize heterogeneity in cooperative MARL. In task allocation settings, our theory shows that the advantage of behavioral diversity is a predictable consequence of reward *curvature*: if the inner aggregator is Schur-convex, amplifying inequality, and the outer aggregator is Schur-concave, amplifying uniformity, heterogeneous policies are strictly superior; reversing the curvature removes the benefit. Complementing this analysis, and covering settings where our theory doesn't apply, the proposed HetGPS algorithm automatically steers underspecified environments to either side of the diversity boundary, letting us encourage or suppress heterogeneity and providing a sandbox for studying its advantages. Together, these results help turn the choice of heterogeneity from an ad-hoc heuristic into a controllable design dimension, and help reconcile past mixed results on parameter sharing.

A key remaining open question concerns how the environment's *transition dynamics* interact with reward curvature to shape heterogeneity gains. We expand on this open question, other directions for future work, and our scope/limitations, in App. R.

### REPRODUCIBILITY STATEMENT

Our supplementary website contains the source code we used to produce all results in this work, including the code used to train the agents, our implementation of HetGPS, and the code used to produce all plots in the paper (see Appendix C). The `readme` contains detailed instructions on how to use this code. All mathematical claims made in this work are fully proven in the Appendix, and our assumptions are described in detail in Section 2 ("Problem Setting"). Appendices Q and P address potential questions readers may have regarding the stability of the bilevel optimization process used in HetGPS and similar environment design algorithms in the literature. Finally, Appendix R addresses the assumptions and scope of our work, and outlines some remaining open questions.

### ACKNOWLEDGMENTS

This work is supported by European Research Council (ERC) Project 949940 (gAIa) and ARL DCIST CRA W911NF-17-2-0181. We gratefully acknowledge their support.

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

## A  COMPUTATIONAL RESOURCES USED

For the realization of this work, we have employed computational resources that have gone towards: experiment design, prototyping, and running final experiment results. Simulation and training are both run on GPUs, no CPU compute has been used. Results have been stored on the WANDB cloud service. We estimate:

- 300 compute hours on an NVIDIA GeForce RTX 2080 Ti GPU.
- 500 compute hours on an NVIDIA L40S GPU.

Simply reprouing our results using the available code will take considerably less compute hours (around a day).

## B  CODE AND DATA AVAILABILITY

We provide the link to the code in the supplementary website. The code contains instructions on how to reproduce the experiments in the paper and dedicated YAML files containing the hyperparameters for each experiment presented. The YAML files are structured according to the HYDRA (Yadan, 2019) framework which allows smooth reproduction as well as systematic and standardized configuration. We further attach all scripts to reproduce the plots in the paper from the experiment results.

## C  IMPLEMENTATION DETAILS

For all experiments, we use the MAPPO MARL algorithm (Yu et al., 2022). Environments are implemented in the multi-agent environment simulator VMAS (Bettini et al., 2022), and trained using TorchRL (Bou et al., 2023). Both the actor and critic are two-layer MLPs with 256 neurons per layer and Tanh activation. Further details, such as hyperparameter choices, are available in the attached code and YAML configuration files.

Experiments were run on a single Nvidia L40 GPU. In the HetGPS experiments (Sec. 4), a standard MARL training iteration (60,000 frames) takes approx. 15s; including the environment backpropagation increases this to approximately 20s.

## D  USE OF LLMS

We used LLMs (ChatGPT 4o and Gemini 2.5 Pro) to improve some parts of the writing, e.g., make wording suggestions. We verified and take responsibility for all LLM-related outputs in this work.

## E  COLONEL BLOTTO & LEVEL-BASED FORAGING

We describe how two well-known settings from the literature fit into our theoretical framework, and check what our theoretical results say about their heterogeneity gain.

### E.1  TEAM COLONEL BLOTTO (FIXED ADVERSARY)

The *Colonel Blotto game* is a well-known allocation game studied in both game theory and MARL (Roberson, 2006; Noel, 2022). It is used to model election strategies and other resource-based competitions. In the team variant with fixed adversary, $N$ friendly colonels (agents) each distribute a (fixed and equal) budget of troops $r_{ij} \geq 0$, $\sum_{j=1}^{M} r_{ij} = 1$ across $M$ battlefields $j \in \{1, \ldots, M\}$ (our tasks). A fixed adversary selects a *stochastic* opposing allocation strategy, i.e. a distribution $\pi_{\mathrm{adv}}$ over vectors $a = (a_1, \ldots, a_M)$ which is fixed throughout training and evaluation. Let $s_j = \sum_{i=1}^{N} r_{ij}$ denote the team force committed to battlefield $j$. Our agents win against the adversary if the troops they allocate to a given field surpass the troops allocated by the adversary. The expected value secured on battlefield $j$ is therefore

$$T_j(a_j) \ = \ v_j \, \mathbb{E}_{a \sim \pi_{\mathrm{adv}}}\big[\mathbb{1}[\,s_j > a_j\,]\big] \ = \ v_j \Pr_{a \sim \pi_{\mathrm{adv}}}\big[s_j > a_j\big],$$

where $1[x > y]$ denotes the indicator function. This is a **thresholded-sum** that remains symmetric and coordinate-wise non-decreasing in every agent's contribution $r_{ij}$. Aggregating across battlefields with a value-weighted sum yields the team reward

$$R(A) \;=\; \sum_{j=1}^{M} T_j(a_j) \;=\; \underbrace{\sum_{j=1}^{M}}_{U} T_j\Big(\underbrace{\sum_{i=1}^{N} r_{ij}}_{\oplus}\Big),$$

so the game fits the double-aggregation structure $R(A) = \bigoplus_j \bigoplus_i r_{ij}$ assumed in our analysis.

Heterogeneity Gain: This is a continuous allocation game, and the inner aggregator $T_j$ is an indicator function over the sum of troop allocations to battlefield $j$. This function is Schur-concave (and Schur-convex at the same time!). Hence, by Thm. 3.2, heterogeneous colonel teams, where each colonel has a distinct troop allocation strategy, have no advantage over homogeneous teams, where all colonels employ the same allocation strategy: $\Delta R = 0$. This makes sense, as it makes no difference whether two different colonels allocate $x/2$ troops to a battlefield, or one colonel allocates $x$ troops to the battlefield.

Our analysis also tells us what happens when we change $T_j$: this provides insights for generalizations of the Colonel Blotto game. For example, maybe the troops of different colonels don't cooperate as well with each other, such that two colonels allocating $x/2$ troops to a battlefield results in a lower $T_j$-value than a single colonel allocating $x$ troops. In this case, $T_j$ becomes strictly Schur-convex, and Thm. 3.1 tells us that $\Delta R > 0$ as long as the optimal allocation is non-trivial. Hence, heterogeneous teams are advantaged.

### E.2   LEVEL-BASED FORAGING

The well-known *level-based foraging* (LBF) benchmark, based on the knapsack problem (Garey et al., 1990), is a deceptively challenging, embodied MARL environment, where $N$ agents are placed on a grid with $M$ food items, and are tasked with collecting them. Each item $j$ has an integer level $L_j$ that must be met or exceeded by the combined skills of the agents standing on that cell before it can be collected (Papoudakis et al., 2021). Let agent $i$'s skill be $e_i$. At a given step the binary variable

$$r_{ij} \;\in\; \{0, e_i\}, \qquad \text{with } \sum_{j=1}^{M} r_{ij} \le e_i,$$

denotes whether $i$ contributes its skill to item $j$. In our setting, we assume all agents are equally skilled, so $e_i = 1 \ \forall i$. Collecting these variables thus yields an allocation matrix $A = [r_{ij}] \in \{0, e_i\}^{N \times M}$, which again matches our framework.

**Inner aggregator.**   A food item is harvested if the summed skill on its cell reaches the threshold, so

$$T_j(a_j) \;=\; L_j \ 1\Big[\sum_{i=1}^{N} r_{ij} \;\ge\; L_j\Big], \qquad a_j = (r_{1j}, \ldots, r_{Nj})^{\top}.$$

This **threshold–sum** is symmetric and monotone, depending only on the sum of its arguments and therefore simultaneously Schur-convex and Schur-concave.

**Outer aggregator.**   The stepwise team reward is the sum of harvested item values,

$$R(A) \;=\; \sum_{j=1}^{M} T_j\Big(\sum_{i=1}^{N} r_{ij}\Big) \;=\; \underbrace{\sum_{j=1}^{M}}_{U} T_j\Big(\underbrace{\sum_{i=1}^{N} r_{ij}}_{\oplus}\Big) \;=\; \bigoplus_{j=1}^{M} \bigoplus_{i=1}^{N} r_{ij},$$

so LBF also conforms to the double-aggregation form $R(A) = \bigoplus_j \bigoplus_i r_{ij}$.

**MARL Environment Reward.** In the level-foraging environment, items that are picked up either disappear; replace themselves with different items; or replace themselves with the same item (possibly at a different cell). In all of these cases we can represent the cumulative reward as $\sum_{t=0}^{T} \gamma^t R_t(A_t)$ for some sequence $(R_t)_{t=1,\dots T}$ of rewards adhering to the above reward structure.

Heterogeneity Gain: We analyze the heterogeneity gap of a specific stepwise reward $R$.

Because this is an embodied environment where each agent can either stand on an item ($r_{ij} = 1$) or not ($r_{ij} = 0$), effort allocations are *discrete*. Our continuous curvature test therefore does not apply directly, but the discrete analysis in Fig. 2 (left panel) does.

The table in Fig. 2 tells us something about the case where all items have level $L_j = 1$. In this case, since we assumed $e_i = 1$ for all agents, the inner aggregator reduces to

$$T_j(a_j) = \mathbb{1}\Big[\sum_{i=1}^{N} r_{ij} \geq 1\Big] = \max_i r_{ij},$$

while the outer aggregator is an unnormalized sum, which becomes the *mean* when divided by $M$. Hence $R(A) = \sum_j \max_i r_{ij}$, which, up to the constant $1/M$, is exactly the case $U = \text{mean}$, $T = \max$ of Fig. 2. That table shows

$$\frac{1}{M}\Delta R \;=\; \frac{\min\{M, N\} - 1}{M},$$

so the heterogeneity gap is *strictly positive* whenever the team could in principle cover more than one item ($\min\{M, N\} > 1$). Intuitively, a homogeneous team can only collect one item per step (all agents flock to the same cell), whereas heterogeneous agents may spread out and capture up to $\min\{M, N\}$ items simultaneously.

This analysis can be extended to the case where all items have the same level $L > 1$ and $L \mid N$ by grouping agents into $\tilde{N} := N/L$ *agent teams*, each bundle contributing exactly $L$ units of skill. This yields

$$\frac{1}{ML}\Delta R \;=\; \frac{\min\{M, \tilde{N}\} - 1}{M}.$$

(We omit the formal analysis, which is not difficult). Thus, if the team can form at least two such bundles ($\tilde{N} > 1$), heterogeneity is again advantageous. If it cannot, then $\Delta R = 0$, and there is no advantage to heterogeneity.

When the levels $\{L_j\}$ differ, an exact closed form is harder, but in general we expect $\Delta R > 0$ whenever there is some combination of items that the heterogeneous team can collect, which in total is worth more than the largest single item that can be collected if all $N$ agents stand on its cell.

In LBF, therefore, our theory suggests that behavioral diversity is often advantageous. Note that (unlike the Colonel Blotto game) since LBF is an embodied, time-extended MARL environment, this analysis does not *formally guarantee* an advantage to RL-based heterogeneous agent teams: rather, it identifies that there are effort allocation strategies that will give these teams an advantage over homogeneous teams. The agents must still *learn* and be able to execute these strategies to gain this advantage (e.g., they must learn how to move to attain the desired allocations).

## F  SUM-FORM AGGREGATORS

Many useful reward functions are *sum-form aggregators*:

**Definition F.1** (Sum-Form Aggregator). *A task-level aggregator $f : \mathbb{R}^N \to \mathbb{R}$ for task $j$ is a **sum-form aggregator** if it can be written as: $f(x_j) = \sum_{i=1}^{N} g(x_j)$, where $g_j : \mathbb{R} \to \mathbb{R}$ is differentiable. We say $f$ is (strictly) convex or concave if $g$ is (strictly) convex or concave, respectively.*

Tab. 3 contains examples. When our aggregators have this form, Schur-convexity (concavity) is determined by whether $g$ is convex (concave)–a simple computational test. This is because of the following *known* connection between sum-form aggregators and Schur-convexity/concavity:

**Lemma F.1** (Schur Properties of Sum-Form Aggregators (Peajcariaac & Tong, 1992))**.** *Given sum-form task-level aggregator $f(x) = \sum_{i=1}^{N} g(x_i)$, the following holds: **(i)** if $g$ is (strictly) convex, then $f$ is (strictly) Schur-convex; and **(ii)** if $g$ is (strictly) concave, then $f$ is (strictly) Schur-concave.*

This lemma simplifies checking the conditions of our heterogeneity gain results. For example, the following corollary can be used to establish $\Delta R > 0$ for many of the aggregators in Tab. 3:

**Corollary F.1** (Convex-Concave Positive Heterogeneity Gain). *Let $N, M \geq 2$. Let $g : [0, 1] \to \mathbb{R}_{\geq 0}$ be a non-negative strictly convex function satisfying $g(0) = 0$, and let $h : \mathbb{R}_{\geq 0} \to \mathbb{R}$ be a strictly concave, increasing function satisfying $h(0) = 0$. If each task-level aggregator is a strictly convex sum-form aggregator $T_j(a_j) = \sum_{i=1}^{N} g(r_{ij})$, and the outer aggregator is a strictly concave sum-form aggregator $U(y) = \sum_{j=1}^{M} h(y_j)$, then $\Delta R > 0$.*

*Proof of Corollary F.1.* We will apply Theorem 3.1 by verifying its conditions:

First, by Lemma F.1, since $g$ is strictly convex, the (identical) task-level aggregators $T_j(x) = \sum_{i=1}^{N} g(x_i)$ are strictly Schur-convex, satisfying condition (i) of Theorem 3.1.

Second, the outer aggregator $U(y_1, \ldots, y_M)$ is strictly increasing at every coordinate by definition, satisfying condition (ii).

Hence, the conditions of Thm. 3.1 apply. To establish $\Delta R > 0$, it remains to verify that the optimal allocation is non-trivial: it distributes effort across at least two tasks. In any admissible *homogeneous* solution, each of the $N$ agents chooses the same effort-distribution $(c_1, \ldots, c_M)$ on tasks, with $\sum_j c_j = 1$. Then task $j$'s reward is $T_j = N g(c_j)$, so $R(A) = \sum_{j=1}^{M} h(N g(c_j))$. The trivial, *all-agent single-task allocation* uses $(c_j = 1, c_{k \neq j} = 0)$. Its reward is therefore $R_{\text{corner}} = h(N g(1)) + \sum_{k \neq j} h(N g(0)) = h(N g(1))$ since $g(0) = 0$ and $h(0) = 0$.

Strict concavity of $h$ implies that $h(N g(1)) < N \cdot h(g(1))$. Hence, agents can attain a better reward by allocating effort 1 to $N$ different tasks rather than a single task. This shows that the best solution *must* use at least two nonzero $c_j$, completing the proof. $\square$

## G  FORMAL ANALYSIS

### G.1  PROOF OF THM. 3.1

*Proof of Thm. 3.1.* Let $A_{\text{hom}}$ be an optimal homogeneous allocation (i.e., $R(A_{hom}) = R_{hom}$), whose $i$th row is the vector

$$c = (c_1, \ldots, c_M) \quad \text{with} \quad \sum_{j=1}^{M} c_j = 1.$$

Then each column $j$ of $A_{\text{hom}}$ is the uniform vector $u_j = (c_j, c_j, \ldots, c_j)^\top \in \mathbb{R}^N$. Hence the task-level reward is $T_j(u_j)$, and the overall reward is

$$R(A_{\text{hom}}) = U(T_1(u_1), \ldots, T_M(u_M)).$$

Because $\sum_j c_j = 1$, there is at least one task $j$ with $c_j > 0$. We construct a heterogeneous allocation $A_{het}$ such that each column $x_j$ in $A_{het}$ has the same sum as the corresponding column in $A_{hom}$.

The total effort allocated to a task $j$ can be expressed as $\lfloor N c_j \rfloor + f_k$, where $0 \leq f_j < 1$. First, we assign $\lfloor N c_j \rfloor$ agents to allocate effort 1 to task $j$, for every task $j$. These agents are all distinct. This leaves us with $\sum_j f_j = N - \sum_j \lfloor N c_j \rfloor$ agents that have not allocated any effort yet. Let $i$ be the first of those agents. We have agent $i$ allocate $f_1$ effort to task 1, $f_2$ effort to task 2, and so on, until we arrive at a task $k$ such that $f_1 + \ldots + f_k = 1 + s$, for some $s > 0$. We have $i$ allocate $f_k - s$ to this task $k$. Then, we move to agent $i + 1$, and allocate the remaining fractional efforts in the same manner (and in particular, allocating $s$ effort to task $k$), until agent $i + 1$ overflows. Then we move to agent $i + 2$, and so on. This ensures that we have allocated $N$ effort in total across the agents, and that every agent's effort allocation sums exactly to 1, so is feasible.

Let $x_j$ be the $j$th column of $A_{het}$. We note the following fact: any non-uniform vector whose sum is $N c_j$ majorizes the uniform vector $u_j$. Hence, $T_j(x_j) \geq T_j(u_j)$, with equality only if $x_j = u_j$. This means that if $A_{het} \neq A_{hom}$, then

$$R\big(A_{\text{het}}\big) \;=\; U\big(T_1(x_1),\ldots,T_M(x_M)\big) \;>\; U\big(T_1(u_1),\ldots,T_M(u_M)\big) \;=\; R\big(A_{\text{hom}}\big).$$

We note that $A_{hom} = A_{het}$ only if $A_{hom}$ is a trivial allocation, as $A_{het}$ contains at least one agent allocating effort 1 to some task, and $A_{hom}$'s agents only allocate fractional efforts, if it is non-trivial. Otherwise, since $R(A_{hom}) = R_{hom}$, the above inequality implies $\Delta R = R_{\text{het}} - R_{\text{hom}} > 0$. This completes the proof. $\qquad\square$

### G.2 PROOF OF THM. 3.2

*Proof of Thm. 3.2.* Let $A$ be an arbitrary feasible allocation, and let $A_{\text{hom}}$ be a *homogeneous* allocation with the same column sums. Concretely, for each column $j$, define

$$s_j \;=\; \sum_{i=1}^{N} r_{ij} \quad \text{and} \quad u_j \;=\; \big(\tfrac{s_j}{N}, \tfrac{s_j}{N}, \ldots, \tfrac{s_j}{N}\big)^{\top},$$

so $u_j$ is the *uniform* distribution of total mass $s_j$ across $N$ agents. Then construct

$$A_{\text{hom}} \;=\; \begin{pmatrix} \frac{s_1}{N} & \cdots & \frac{s_M}{N} \\ \vdots & \ddots & \vdots \\ \frac{s_1}{N} & \cdots & \frac{s_M}{N} \end{pmatrix},$$

which is clearly *homogeneous* (each row is the same), and respects each column sum $s_j$. Since $\sum_j s_j = N$, each row sums to 1, hence the allocation is feasible. By Schur-concavity of $T_j$, for each column $j$ we have

$$a_j \;\succ\; u_j \quad \implies \quad T_j(a_j) \;\leq\; T_j(u_j),$$

unless $a_j$ is $u_j$. In other words, *any* deviation from the uniform vector with the same sum $\sum_{i=1}^{N} a_{ji} = s_j$ will not increase $T_j(a_j)$ under Schur-concavity. Hence for each column $j$ of $A$, $T_j(a_j) \leq T_j(u_j)$,. Since $U$ is non-decreasing in each coordinate,

$$R\big(A\big) \;=\; U\big(T_1(a_1),\ldots,T_M(a_M)\big) \;\leq\; U\big(T_1(u_1),\ldots,T_M(u_M)\big) \;=\; R\big(A_{\text{hom}}\big).$$

This implies $\Delta R \;=\; 0.$ $\qquad\square$

### G.3 PROOF OF THM. 3.3

*Proof of Thm. 3.3.* By hypothesis, the components of the task score vector

$$\mathbf{T}(\mathbf{A}) = \Big(T_1(a_1),\, T_2(a_2),\, \ldots,\, T_M(a_M)\Big)$$

always sum to $C$. By strict Schur-convexity, the maximum value of $U$ over such vectors is attained precisely at an extreme point of the $C$-simplex, i.e. at some permutation of $(C, 0, \ldots, 0)$. Hence, we seek to find an allocation of efforts, $A_{\text{corner}}$, that causes $\mathbf{T}(\mathbf{A})$ to equal this vector.

Let each agent $i$ invest *all* of its effort into task 1. This is the trivial allocation. Then the first column of $A_{\text{corner}}$ is $(1, 1, \ldots, 1)^{\top}$, and all other columns $a_j$ are zero. Since task scores sum to $C$, we get $T_1\big(a_1\big) = C, \quad T_j\big(a_j\big) = 0$ for $j \neq 1$. By assumption (2), we infer that the vector of task-level scores is indeed $(C, 0, \ldots, 0)$.

Notice that *each row of $A_{corner}$ is the same* $(1, 0, \ldots, 0)$, making $A_{\text{corner}}$ a *homogeneous* allocation. Hence, we attained the maximum possible reward $R(\mathbf{A})$ through a homogeneous allocation, implying $\Delta R = 0$. $\qquad\square$

### G.4 PROOF OF THM. 3.4

Before proving the statement, let's write the expressions for homogeneous and heterogeneous optima. For each task $j$, we defined

$$T_j(A) = \sum_{i=1}^{N} \frac{\exp\big(t \cdot r_{ij}\big)}{\sum_{\ell=1}^{N} \exp\big(t \cdot r_{\ell j}\big)} \, r_{ij},$$

while defining the outer aggregator to be

$$U\big(T_1(a_1), \ldots T_M(a_m)\big) = \sum_{j=1}^{M} \frac{\exp\big(\tau \cdot T_j(A)\big)}{\sum_{\ell=1}^{M} \exp\big(\tau \cdot T_\ell(A)\big)} \, T_j(A),$$

where $t, \tau \in \mathbb{R}$ are temperature parameters. In the **homogeneous setting**, where all agents share the same allocation $c = (c_1, \ldots, c_M)$, we therefore have $T_j(A) = \sum_{i=1}^{N} \frac{\exp\big(t\, c_j\big)}{\sum_{\ell=1}^{N} \exp\big(t\, c_j\big)} \, c_j = c_j$. Thus,

$$R_{\text{hom}} = \max_{c \in \Delta^{M-1}} \quad \sum_{j=1}^{M} \frac{\exp\big(\tau\, c_j\big)}{\sum_{\ell=1}^{M} \exp\big(\tau\, c_\ell\big)} \, c_j$$

where $\Delta^{M-1}$ is the simplex of all admissible allocations.

In the general **heterogeneous setting**, each row $(r_{i1}, \ldots, r_{iM})$ can be different. Then

$$T_j(A) \;=\; \sum_{i=1}^{N} \frac{\exp\big(t\, r_{ij}\big)}{\sum_{\ell=1}^{N} \exp\big(t\, r_{\ell j}\big)} \, r_{ij},$$

and we choose $A \in (\Delta^{M-1})^N$ to maximize

$$R_{\text{het}} \;=\; \max_{A} \sum_{j=1}^{M} \frac{\exp\big(\tau\, T_j(A)\big)}{\sum_{k=1}^{M} \exp\big(\tau\, T_k(A)\big)} \, T_j(A).$$

Keeping these expressions in mind, we proceed with the proof of Thm. 3.4.

Reminder: assuming $N = M \geq 2$, we want to prove $\Delta R(t, \tau; N) = 0$ when $t \leq 0$, and

$$\Delta R(t, \tau; N) \geq \begin{cases} \sigma(t, N) - \dfrac{1}{N}, & t > 0, \ \tau \leq 0, \\[2mm] \max\big\{\sigma(t, N) - \sigma(\tau, N), \, 0\big\}, & t > 0, \ \tau \geq 0. \end{cases}$$

otherwise, where $\sigma(t, N) := \frac{e^t}{e^t + N - 1}$.

*Proof of Thm. 3.4.* When $t \leq 0$, $T_j$ is Schur–concave, so $\Delta R = 0$ by Thm. 3.2. We assume $t > 0$ for the rest of the proof.

*Homogeneous optimum.* If every row of $A$ equals the same allocation $c \in \Delta^{N-1}$, then $T_j(A) = c_j$. $U$ is Schur–concave for $\tau \leq 0$, and Schur–convex for $\tau \geq 0$, hence it is maximized by the uniform distribution in the former case, and by a 1-hot vector in the latter case, yielding:

$$R_{\text{hom}} \;=\; \max_{c \in \Delta^{N-1}} U(c) \;=\; \begin{cases} \dfrac{1}{N}, & \tau \leq 0, \\[2mm] \sigma(\tau, N), & \tau > 0. \end{cases} \tag{H}$$

*Lower bound on $R_{het}$.* The *trivial* allocation, where every agent works on the same task, produces $R_{\text{trivial}} = \sigma(\tau, N)$. The *spread* allocation, where agent $i$ works exclusively on task $i$, makes each column "one-hot"; this gives $T_j = \sigma(t, N)$ for all $j$, and plugging this into $U$, we get $R_{\text{spread}} = \sigma(t, N)$. Consequently

$$R_{\text{het}} \;\geq\; \max\{\sigma(t, N), \, \sigma(\tau, N)\}. \tag{L}$$

Combining (H) and (L) gives the desired lower bound. $\qquad\square$

Table 1: All nine extreme cases of inner/outer aggregators belonging to the set $\{\min, \text{mean}, \max\}$. In each cell, we show the best possible outcome for Heterogeneous vs. Homogeneous allocations and the resulting $\Delta R$.

| | $T = \min$ | $T = \text{mean}$ | $T = \max$ |
|---|---|---|---|
| $U = \min$ | **Inner:** $T_j = \min_i r_{ij}$. **Best** $R_{het}$, $R_{hom}$: All must have $r_{ij} \geq x$ to push $\min_i r_{ij} = x$, so $x \leq 1/M$. $\implies T_j = 1/M$. **Outer:** $\min_j T_j = 1/M \implies R = 1/M$. **Gap:** 0. | **Inner:** $T_j = \frac{1}{N}\sum_i r_{ij}$ (avg over $i$). **Maximize** $\min_j T_j$: Both $R_{het}$, $R_{hom}$ must make $T_j$ all equal (for best min), so $T_j = 1/M$. **Outer:** $\min_j T_j = 1/M \implies R = 1/M$. **Gap:** 0. | **Inner:** $T_j = \max_i r_{ij}$. **Outer:** picks $\min_j T_j$. $R_{het}$: $\min_j T_j = 1 \implies R = 1$. $R_{hom}$: $\min_j T_j = 1/M \implies R = 1/M$. **Gap:** $1 - \frac{1}{M} = \frac{M-1}{M}$. |
| $U = \text{mean}$ | **Inner:** $T_j = \min_i r_{ij} = 1/M$. **Outer:** simple avg $\frac{1}{M}\sum_j T_j$. Since $\sum_j T_j = M \cdot (1/M) = 1 \implies R = 1/M$. Both $R_{het}$, $R_{hom}$ same $\implies \Delta R = 0$. **Gap:** 0. | **Inner:** $T_j = \frac{1}{N}\sum_i r_{ij}$. Then $\sum_j T_j = 1$. **Outer:** avg$= \frac{1}{M}\sum_j T_j$. Hence $R = \frac{1}{M} \cdot 1 = \frac{1}{M}$. Same for $R_{het}$, $R_{hom}$. **Gap:** 0. | **Inner:** $T_j = \max_i r_{ij}$. **Outer:** avg$= \frac{1}{M}\sum_j T_j$. $R_{het}$: sum $= M \implies R = 1$. $R_{hom}$: sum $= 1 \implies R = 1/M$. **Gap:** $1 - \frac{1}{M} = \frac{M-1}{M}$. |
| $U = \max$ | **Inner:** $T_j = \min_i r_{ij}$ can be made 1 for one task. **Outer:** picks $\max_j T_j = 1 \implies R = 1$ Same for $R_{het}$, $R_{hom}$. **Gap:** 0. | **Inner:** $T_j = $ avg over $i$. **Outer:** picks $\max_j T_j$. Both $R_{het}$, $R_{hom}$ can put all effort into one task to get $T_j = 1$, so $R = 1$. **Gap:** 0. | **Inner:** $T_j = \max_i r_{ij}$. **Outer:** picks $\max_j T_j$. Both $R_{het}$, $R_{hom}$ can achieve $\max_j = 1 \implies R = 1$. **Gap:** 0. |

Table 2: A "9 extreme cases" table for *discrete, one-task-per-agent* allocations.

| | min | mean | max |
|---|---|---|---|
| min | **Inner:** $T_j \to \begin{cases} 1, & \text{if all agents pick } j, \\ 0, & \text{otherwise.} \end{cases}$ **Outer:** $\min_j T_j$. To get $R > 0$, must have $T_j > 0$ for *every* $j$ (i.e. all agents pick *all* tasks, impossible). Hence $R_{het} = R_{hom} = 0$ typically, $\Delta R = 0$. | **Inner:** $T_j = \frac{|\mathcal{I}_j|}{N}$ **Outer:** $\min_j T_j$. $R_{het} = \lfloor N/M \rfloor / N$. $R_{hom} = 0$. $\Delta R = \lfloor N/M \rfloor / N$. | **Inner:** $T_j \to \begin{cases} 1, & \text{if at least 1 agent picks } j, \\ 0, & \text{if no agent picks } j. \end{cases}$ **Outer:** $\min_j T_j$. - *Heterogeneous* can choose $s$ distinct tasks. If want $\min_j = 1$, must pick *all* $M$ tasks. That requires $N \geq M$. Then $R = 1$. - *Homogeneous* covers only 1 task $\implies \min_j = 0$ for $M > 1 \implies R = 0$. $\Delta R = 1$ if $N \geq M$, else 0. |
| mean | **Inner:** $T_j = 1$ only if all pick $j$, else 0. Summation $\sum_j T_j$ is number of tasks chosen by *all* agents. Usually 0 or 1. **Outer:** Average across $j$. $R = \frac{1}{M}\sum_j T_j$. $\implies R = 1/M, \Delta R = 0$. | **Inner:** $T_j = \frac{|\mathcal{I}_j|}{N}$. **Outer:** Average across tasks: $R = \frac{1}{M}\sum_{j=1}^{M} \frac{|\mathcal{I}_j|}{N} = \frac{1}{M}$. No matter how agents are distributed, $\sum_{j=1}^{M} |\mathcal{I}_j| = N$. Hence $R_{het} = R_{hom} = \frac{1}{M}$, $\Delta R = 0$. | **Inner:** $T_j = 1$ if chosen by at least 1 agent, else 0. **Outer:** Average across $j$: $\frac{1}{M}\sum_j T_j$. This is $\frac{1}{M} \cdot (\text{\# of tasks chosen})$. - *Heterogeneous* can pick up to $\min(M, N)$ tasks, so $R = \frac{\min(M,N)}{M}$. - *Homogeneous* covers exactly 1 task $\implies R = 1/M$. $\Delta R = \frac{\min(M,N)-1}{M}$. |
| max | **Inner:** $T_j = 1$ only if all pick $j$, else 0. **Outer:** $\max_j T_j$. $\Delta R = 0$. | **Inner:** $T_j = |\mathcal{I}_j|/N$. **Outer** ($\tau \to +\infty$): $\max_j T_j$. We can place *all* agents on one task, get $T_j = 1$. Then $R = 1$. Same for homogeneous or heterogeneous. $\Delta R = 0$. | **Inner:** $T_j = 1$ if at least 1 picks $j$, else 0. **Outer:** $\max_j T_j = 1$ if any agent picks $j$. Even a single task yields $R = 1$. So $R_{hom} = R_{het} = 1$, $\Delta R = 0$. |

# H  DERIVING THE $\{\min, \text{mean}, \max\}$ HETEROGENEITY GAINS IN THE FIG. 2 TABLE

We derive these heterogeneity gain case-by-case. Tab. 1 summarizes the derivation for continuous allocations ($r_{ij} \in [0, 1]$), and Tab. 2 does the same for discrete effort allocations ($r_{ij} \in \{0, 1\}$).

# I   Parametrized Families of Aggregators

The Table in this section illustrates several families of *generalized aggregators* that the analysis in this paper applies to. The scalar $t$ parametrizes each family of aggregators, continuously shifting the aggregators from Schur-concave to Schur-convex.

Table 3: Illustrative families of parametric (and one nonparametric) aggregators $f_t(x)$. Changing the real parameter $t$ can switch between Schur-convex and Schur-concave behaviors (on nonnegative inputs), or control how strongly the aggregator favors "peaked" vs. "uniform" distributions. As $t \to \pm\infty$ or $t \to 0$, many reduce to well-known extremes such as $\max$, $\min$, or the arithmetic mean.

| Name | Definition | Schur Property & Limits |
|---|---|---|
| **Power-Sum** | $$f_t(x) \;=\; \sum_{i=1}^{N} (x_i)^t, \quad x_i \geq 0, \quad t > 0$$ | • Strictly *Schur-convex* for $t > 1$.
• Strictly *Schur-concave* for $0 < t < 1$.
• At $t = 1$, it is linear (both Schur-convex and Schur-concave).
• Undefined at $t \leq 0$ if any $x_i = 0$, though one can extend with limits. |
| **Power-Mean** | $$M_t(x) \;=\; \left( \frac{1}{N} \sum_{i=1}^{N} (x_i)^t \right)^{1/t}, \quad x_i \geq 0, \ t \neq 0$$ | • Strictly *Schur-convex* for $t > 1$.
• Strictly *Schur-concave* for $0 < t < 1$.
• Reduces to arithmetic mean at $t = 1$.
• As $t \to \infty$, converges to $\max_i x_i$; as $t \to -\infty$, converges to $\min_i x_i$. |
| **Log-Sum-Exp (LSE)** | $$\mathrm{LSE}_t(x) \;=\; \frac{1}{t} \ln\!\left( \sum_{i=1}^{N} e^{\,t\,x_i} \right), \quad t \neq 0$$ | • Strictly *Schur-convex* for $t > 0$.
• Strictly *Schur-concave* for $t < 0$.
• As $t \to \infty$, approaches $\max_i x_i$; as $t \to -\infty$, approaches $\min_i x_i$. |
| **Softmax Aggregator** | $$\mathrm{Softmax}_t(x) \;=\; \sum_{i=1}^{N} \frac{e^{\,t\,x_i}}{\sum_{j=1}^{N} e^{\,t\,x_j}}\, x_i, \quad t \in \mathbb{R}$$ | • Strictly *Schur-convex* for $t > 0$.
• Strictly *Schur-concave* for $t < 0$.
• As $t \to \infty$, converges to $\max_i x_i$; as $t \to -\infty$, converges to $\min_i x_i$.
• At $t = 0$, each weight is $\frac{1}{N}$, so $\mathrm{Softmax}_0(x) = \frac{1}{N} \sum_i x_i$. |

# J   Additional Results in the Multi-Agent Multi-Task Matrix Game

We report further details and results on the heterogeneity gains obtained in the multi-agent multi-task matrix game.

## J.1   Game formulation

In Tab. 4 we provide an example of the pay-off matrix in this game for $N = M = 3$.

## J.2   $N = M = 2$

We train with $N = 2, M = 2$ for 100 training iterations (each consisting of 60,000 frames). We report the results for the **continuous** case in Tab. 5 and for the **discrete** case in Tab. 6. The evolution of the heterogeneity gains over training is shown in Fig. 6.

Table 4: Example of a Multi-Agent Multi-Task matrix game for $N = M = 3$. Agents choose their actions $A = (r_{ij})$ and receive the global reward $R(A) = \bigoplus_{j=1}^{M} \bigoplus_{i=1}^{N} r_{ij}$.

|        |   | Tasks |   |   |
|--------|---|-------|---|---|
|        |   | 1 | 2 | 3 |
| Agents | 1 | $r_{11}$ | $r_{12}$ | $r_{13}$ |
|        | 2 | $r_{21}$ | $r_{22}$ | $r_{23}$ |
|        | 3 | $r_{31}$ | $r_{32}$ | $r_{33}$ |

Table 5: Heterogeneity gain $\Delta R \in \mathbb{R}_{0 \leqslant x \leqslant 1}$ of the **continuous** matrix game with $N = M = 2$. The results match the theoretical analysis in the Table of Fig. 2. We report mean and standard deviation after 6 million training frames over 9 different random seeds.

|   |      | $T$ |   |   |
|---|------|-----|---|---|
|   |      | Min | Mean | Max |
|   | Min  | $-0.002 \pm 0.002$ | $0.000 \pm 0.003$ | $\mathbf{0.504 \pm 0.007}$ |
| $U$ | Mean | $-0.002 \pm 0.002$ | $0.000 \pm 0.000$ | $\mathbf{0.496 \pm 0.001}$ |
|   | Max  | $-0.003 \pm 0.002$ | $-0.001 \pm 0.001$ | $0.003 \pm 0.001$ |

Table 6: Heterogeneity gain $\Delta R \in \mathbb{R}_{0 \leqslant x \leqslant 1}$ of the **discrete** matrix game with $N = M = 2$. The results match the theoretical analysis in Fig. 2. We report mean and standard deviation after 6 million training frames over 9 different random seeds.

|   |      | $T$ |   |   |
|---|------|-----|---|---|
|   |      | Min | Mean | Max |
|   | Min  | $0.0 \pm 0.0$ | $\mathbf{0.5 \pm 0.0}$ | $1 \pm 0.0$ |
| $U$ | Mean | $0.0 \pm 0.0$ | $0.0 \pm 0.0$ | $\mathbf{0.5 \pm 0.0}$ |
|   | Max  | $0.0 \pm 0.0$ | $0.0 \pm 0.0$ | $0.0 \pm 0.0$ |

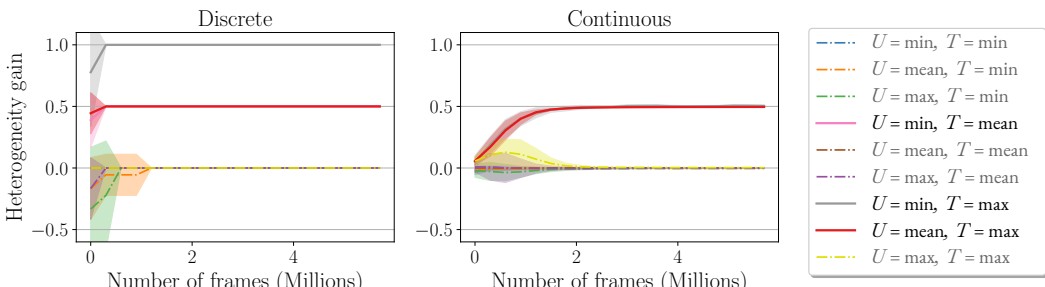

Figure 6: Heterogeneity gain for the discrete and continuous matrix games with $N = M = 2$ over training iterations. We report mean and standard deviation after 6 million training frames over 9 different random seeds. The final results match the theoretical predictions in Fig. 2.

### J.3 $N = M = 4$

In the case $N = M = 4$, the evolution of the heterogeneity gains during training is shown in Fig. 3. We further report the final obtained gains for the **continuous** case in Tab. 7 and for the **discrete** case in Tab. 8.

### J.4 $N = M = 8$ AND $N = 12, M = 2$

To test scalability, we further report results for the discrete matrix game with $N = M = 8$ in Tab. 9, $N = 11, M = 2$ in Tab. 10. In the case of $N = 11, M = 2$, results match the predictions of Table 2 (discrete rewards) precisely.

Table 7: Heterogeneity gain $\Delta R \in \mathbb{R}_{0 \leqslant x \leqslant 1}$ of the **continuous** matrix game with $N = M = 4$. The results match the theoretical analysis in Fig. 2. We report mean and standard deviation after 12 million training frames over 9 different random seeds.

|   |   | $T$ | | |
|---|---|---|---|---|
|   |   | Min | Mean | Max |
|   | Min | $-0.003 \pm 0.002$ | $0.000 \pm 0.001$ | $\mathbf{0.690} \pm 0.026$ |
| $U$ | Mean | $-0.002 \pm 0.000$ | $0.000 \pm 0.000$ | $\mathbf{0.722} \pm 0.002$ |
|   | Max | $-0.037 \pm 0.023$ | $-0.009 \pm 0.005$ | $0.029 \pm 0.006$ |

Table 8: Heterogeneity gain $\Delta R \in \mathbb{R}_{0 \leqslant x \leqslant 1}$ of the **discrete** matrix game with $N = M = 4$. The results match the theoretical analysis in the Table of Fig. 2. We report mean and standard deviation after 12 million training frames over 9 different random seeds.

|   |   | $T$ | | |
|---|---|---|---|---|
|   |   | Min | Mean | Max |
|   | Min | $0.0 \pm 0.0$ | $\mathbf{0.25} \pm 0.0$ | $\mathbf{1.0} \pm 0.0$ |
| $U$ | Mean | $0.0 \pm 0.0$ | $0.0 \pm 0.0$ | $\mathbf{0.75} \pm 0.0$ |
|   | Max | $0.0 \pm 0.0$ | $0.0 \pm 0.0$ | $0.0 \pm 0.0$ |

For $N = M = 8$, we match the predictions precisely except for two cells which have negative empirical $\Delta R$. As discussed in Sec. 5, due to the larger agent scale and action dimensionality (8 agents and 8 tasks), for some reward structures the empirical heterogeneity gain is negative, since the neurally heterogeneous agents we train require more time to discover the optimal policy, hence lag behind their homogeneous counterparts under a fixed training budget. Increasing the number of training steps steers the negative $\Delta R$ values to 0. This nuance aside, what is important is that the results still follow our theory exactly in terms of which reward structures yield a *positive* heterogeneity gain. For such reward structures, our theory also precisely predicts the numerical value of $\Delta R$ (Fig. 2).

Table 9: Heterogeneity gain $\Delta R \in \mathbb{R}_{0 \leqslant x \leqslant 1}$ of the **discrete** matrix game with $N = M = 8$. We report mean and standard deviation after 12 million training frames over 8 different random seeds.

|   |   | $T$ | | |
|---|---|---|---|---|
|   |   | Min | Mean | Max |
|   | Min | $0.0 \pm 0.0$ | $\mathbf{0.125} \pm 0.0$ | $\mathbf{1.0} \pm 0.0$ |
| $U$ | Mean | $-0.094 \pm 0.068$ | $0.0 \pm 0.0$ | $\mathbf{0.875} \pm 0.0$ |
|   | Max | $-0.75 \pm 0.5$ | $0.0 \pm 0.0$ | $0.0 \pm 0.0$ |

Table 10: Heterogeneity gain $\Delta R \in \mathbb{R}_{0 \leqslant x \leqslant 1}$ of the **discrete** matrix game with $N = 11, M = 2$. We report mean and standard deviation after 12 million training frames over 8 different random seeds. The results match the predictions of Table 2 (discrete rewards) precisely.

|   |   | $T$ | | |
|---|---|---|---|---|
|   |   | Min | Mean | Max |
|   | Min | $0.0 \pm 0.0$ | $\mathbf{0.454545 \approx 5/11} \pm 0.0$ | $\mathbf{1.0} \pm 0.0$ |
| $U$ | Mean | $-0.094 \pm 0.0$ | $0.0 \pm 0.0$ | $\mathbf{0.5} \pm 0.0$ |
|   | Max | $-0.75 \pm 0.5$ | $0.0 \pm 0.0$ | $0.0 \pm 0.0$ |

## K   MULTI-GOAL-CAPTURE

In Multi-goal-capture, agents need to navigate to goals. Each agent observes the relative position to the goals, and agent actions are continuous 2D forces that determine their direction of motion. The entries $r_{ij}^t$ of matrix $A^t$ at time $t$ represent the local reward of agent $i$ towards goal $j$, computed as

$r_{ij}^t = \left(1 - d_{ij}^t / \sum_{j=1}^{M} d_{ij}^t\right) / (M-1)$, where $d_{ij}^t$ is the distance between agent $i$ and goal $j$. This makes it so that $\sum_{j=1}^{M} r_{ij}^t = 1$ and $r_{ij}^t \geqslant 0$. At each step, the agents receive the global reward $R(A^t)$, with aggregators $U, T \in \{\min, \mathrm{mean}, \max\}$.

Our results, shown in Fig. 4a and Tab. 11, show that our curvature theory reliably predicts when there is a heterogeneity gain (Fig. 2): it is positive *only* for the concave–convex pairs $U = \min$, $T = \max$ and $U = \mathrm{mean}$, $T = \max$. The heterogeneity gain is smaller in the latter case because learning dynamics matter: with $U = \min$, $T = \max$ the best homogeneous policy is unique (every agent must steer to the midpoint between the two goals) so homogeneous learners seldom find it, leaving room for heterogeneous policies to excel (see App. K). By contrast, $U = \mathrm{mean}$, $T = \max$ admits a continuum of good homogeneous policies, which homogeneous teams execute more easily. For $U = \max$, $T = \min$ and $U = \max$, $T = \mathrm{mean}$, the theoretical $\Delta R$ is 0, yet the empirical heterogeneity gap is negative (Fig. 4a). This occurs because the reward peaks only when all agents coordinate on the same goal. Neurally heterogeneous teams learn this uniform behavior slower than homogeneous teams, so they underperform within the fixed training budget. Additional training would close this gap to $\Delta R = 0$.

Table 11: Heterogeneity gain at the end of training for the Multi-goal-capture experiments in Fig. 4a. We report mean and standard deviation after 30 million training frames over 9 different random seeds.

|  |  | | $T$ | |
|---|---|---|---|---|
|  |  | Min | Mean | Max |
|  | Min | $0.0 \pm 0.02$ | $0.01 \pm 0.02$ | $\mathbf{0.21} \pm 0.03$ |
| $U$ | Mean | $-0.03 \pm 0.01$ | $0.0 \pm 0.0$ | $\mathbf{0.12} \pm 0.09$ |
|  | Max | $-0.08 \pm 0.06$ | $-0.07 \pm 0.08$ | $0.0 \pm 0.0$ |

In Fig. 7 we juxtapose two representative $N = M = 2$ roll-outs of the MULTI-GOAL-CAPTURE environment for *homogeneous* teams (top row) and *heterogeneous* teams (bottom row) when $U = \min$, $T = \max$. Consistent with the discussion in Sec. 5, homogeneous agents steer to the geometric midpoint between the two goals, producing almost overlapping paths–this is suboptimal, as they cannot cover both goals. On the other hand, heterogeneous agents exaggerate their differences, taking sharply diverging trajectories and ensuring one goal each.

## L   2V2 TAG EXPERIMENTS

The goal of our tag experiment is to showcase that our theoretical results, which predict the value of $\Delta R$ based on the curvature of the aggregators, hold for discrete, sparse rewards. Specifically, our results for discrete efforts in Fig. 2 predict that only $(U, T) = (\min, \max), (\min, \mathrm{mean}), (\mathrm{mean}, \max)$ will have positive heterogeneity gain, with $(\min, \max)$ maximizing the gain. We show in Fig. 4b and Tab. 12 that this holds in 2v2 tag, despite the fact that this is a challenging, embodied, long-horizon, whereas our formal results are for instantaneous allocation games[4]. Note that this is a highly interpretable result: a $(\min, \max)$ means that agents are only awarded when *both* escapers are caught, incentivizing heterogeneous strategies where chaser agents split their behaviour so that the chasing efforts $r_{ij}^t$ are equally distributed between both escapers. Fig. 8 visualizes the trajectories learned by agents trained under this $(\min, \max)$ reward structure, showing distinct emergent pursuit strategies emerging depending on whether the agents are neurally heterogeneous or neurally homogeneous.

To test the robustness of our predictions to greater number of agents, we also ran an experiment with 11 agents: 8 chasers and 3 escapers. We trained the agents over 500 episodes of length 1000 each (this episode length is more than twice as long as our other experiments, indicating that our predictions are stable over longer horizons). Note that we still collect the same amount of total frames (30M) as we reduce the number of environments sampled in parallel.

---

[4]The gain for $(\mathrm{mean}, \max)$ is small compared to the other two aggregator combinations, but still positive at $\Delta R \approx 0.37$. This is also significantly higher than aggregator combinations for which we predict $\Delta R$ is not positive, the largest of which attained $\Delta R < 0.01$.

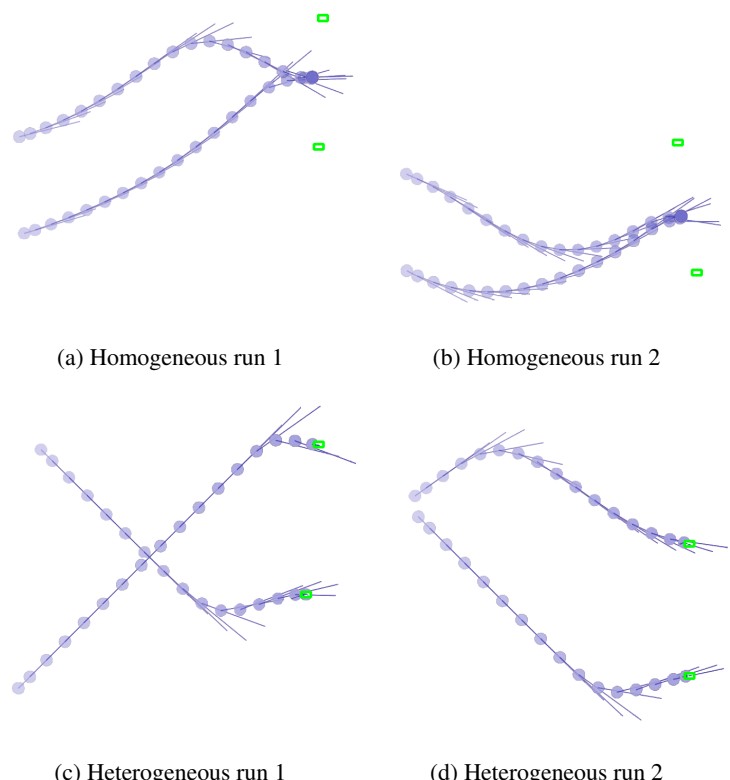

(a) Homogeneous run 1                    (b) Homogeneous run 2

(c) Heterogeneous run 1                   (d) Heterogeneous run 2

Figure 7: **Behaviour under the concave–convex aggregator** $U = \min, T = \max$**.** Each dot is an agent position; line segments indicate instantaneous velocity; green squares mark goal locations. Homogeneous policies collapse to a single "mid-point" route, while heterogeneous policies split and follow distinct paths to cover both goals. Note how the heterogeneous agents *exaggerate* the difference in their trajectories, rather than head directly to the goal: this is an outcome of the reward structure, which encourages maximal diversity.

Due to the high computational cost associated with these experiments, we selected 2 aggregator combinations for which Table 2 predicts a positive $\Delta R$: $(U, T) = (\min, \text{mean})$ and $(U, T) = (\min, \max)$, and we further selected three "control" combinations for which we expect $\Delta R \leq 0$. The results, shown in Figure 9 (discrete rewards), illustrate that our predictions still hold in this case. The final $\Delta R$ values are $(U = \min, T = \text{mean}) = 1.243 \pm 0.615$, $(U = \min, T = \max) = 0.112 \pm 0.084$, $(U = \text{mean}, T = \text{mean}) = -0.196 \pm 0.053$, $(U = \text{mean}, T = \max) = -0.990 \pm 1.025$, and $(U = \max, T = \max) = -3.709 \pm 2.491$.

It is important to note that, while our theoretical predictions regarding when $\Delta R > 0$ hold for any number of agents $N$, they specifically tell us what happens when agents allocate their efforts optimally. The empirical heterogeneity gain $\Delta R$ crucially depends on the quality of strategy agents learn in practice, which in turn also depends on the transition dynamics of the environment, complicating things. When the number of agents or complexity of the task is very large, we may eventually witness a divergence between the empirical heterogeneity gain and the theoretical predictions, for this reason. This does not indicate a problem with our theoretical predictions. Rather, it is a limitation of learning-based methods; using better methods in environments which enable optimal effort allocation will lead to empirical results that more closely mirror our predictions.

## M  FOOTBALL EXPERIMENTS

In some environments, the reward structure might not entirely follow the double-generalized-aggregator structure we study in this work, but at least some part of the reward function might

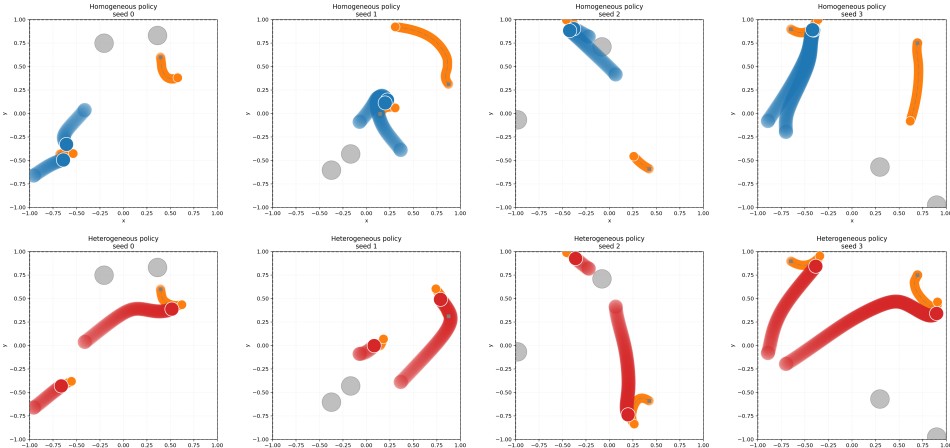

Figure 8: Comparison of homogeneous (top row) and heterogeneous (bottom row) 2v2 tag policies for chaser agents, trained with the reward structure $U = \min, T = \max$ across different initializations. Every column shows the trajectory of the homogeneous (top) and heterogeneous (bottom) policies. (Note that trajectories here are smoothened; agents don't go over obstacles in actual execution). The heterogeneous policies prioritize capturing both agents, whereas the homogeneous policies focus on just one. In the $U = \min, T = \max$ setting, this gives heterogeneous agents greater reward, hence $\Delta R > 0$. Please find more visualizations on our website.

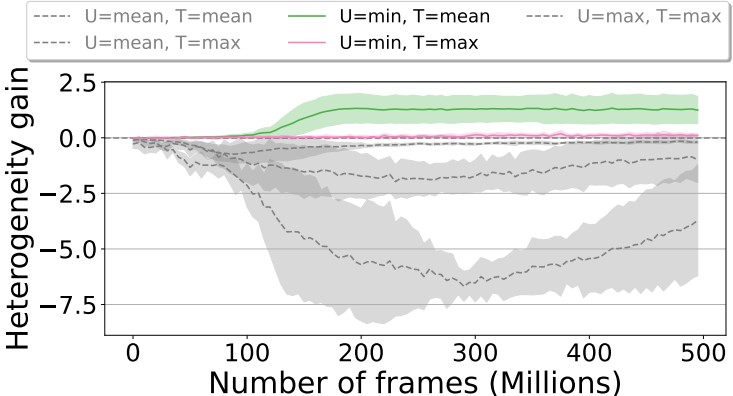

Figure 9: Heterogeneity gain for 8v3 Tag throughout training. We report mean and standard deviation for 30 million training frames over 8 random seeds. Positive aggregator combinations are colored and follow the predictions of Table 2 for discrete rewards. Final gain values are reported in the text of Appendix L.

obey this structure. In our study of the VMAS football scenario (Bettini et al., 2022), we ask what happens when this is the case.

Football is a complex, embodied, long-horizon scenario that requires the agents to learn low-level dribbling skills as well as high-level strategy purely from a shared cooperative reward. The VMAS

Table 12: Heterogeneity gain at the end of training for the Tag experiments in Fig. 4b. We report mean and standard deviation after 30 million training frames over 9 different random seeds.

|  |  | | $T$ | |
|  |  | Min | Mean | Max |
|---|---|---|---|---|
|  | Min | $0.0 \pm 0.0$ | $\mathbf{1.68} \pm 0.24$ | $\mathbf{3.47} \pm 0.23$ |
| $U$ | Mean | $-0.03 \pm 0.04$ | $-0.02 \pm 0.06$ | $\mathbf{0.36} \pm 1.44$ |
|  | Max | $-0.02 \pm 0.09$ | $-0.11 \pm 0.18$ | $-0.30 \pm 0.15$ |

Table 13: Football heterogeneity gains across different reward formulations. Results obtained after 500 training iterations of 240k frames each (6 seeds). Opponent speed annealed from 0% to 100%.

| Reward | $\Delta R$ | Theory $\Delta R > 0$? | Reward meaning |
|---|---|---|---|
| $U = \min,\ T = \max$ | $\mathbf{1.76 \pm 0.72}$ | Yes | One agent should attend the ball, the other the opponent; reward capped by the less-covered task. |
| $U = \text{mean},\ T = \max$ | $\mathbf{1.18 \pm 0.11}$ | Yes | Similar to $(\min, \max)$, but reward is dictated by average task performance. |
| $U = \text{mean},\ T = \text{mean}$ | $0.01 \pm 0.07$ | No | Agents should attend both the opponent and the ball. |
| $U = \min,\ T = \min$ | $-0.08 \pm 0.73$ | No | At least one agent should attend at least the opponent or the ball. |

scenario uses reward shaping to enable agents to learn such behaviors. We *add* a reward structure $U(T(r_{11}^t, r_{21}^t), T(r_{12}^t, r_{22}^t))$ on top of this and ask how this affects heterogeneity.

In our experimental scenario, two learning agents spawn at midfield. A ball is located between them and the goal to the right; a heuristic defender spawns to their left and chases the ball. Agents receive a global reward that increases when the ball moves toward the goal and the defender stays away from it. Additionally, we reuse the reward structure from our Multi-Goal-Capture to define rewards for two tasks: tackling the ball, and tackling the opponent.

The effort at time $t$ is:

$$r_{ij}^t = (1 - \frac{d_{ij}^t}{\sum_j d_{ij}^t})/d_{ij}^t,$$

where $d_{ij}$ is distance of agent $i$ to ball or opponent). The global reward given to all agents is then computed as:

$$R^t = U(T(r_{11}^t, r_{21}^t), T(r_{12}^t, r_{22}^t)) + \beta[(d_{ball,goal}^{t-1} - d_{ball,goal}^t) - (d_{opp,ball}^{t-1} - d_{opp,ball}^t)],$$

where $\beta$ weighs the global football reward.

Since this reward structure does not follow our theory entirely, we ask whether, when $U, T$ are, respectively, strictly Schur-concave and strictly Schur-convex, we should expect $\Delta R > 0$ as in our other scenarios. We test this for $U = \min, T = \max$ and $U = \text{mean}, T = \max$. To control for the possibility that football is heterogeneous "by default", we also test the aggregator combinations $U = \text{mean}, T = \text{mean}$ and $U = \min, T = \min$ as controls.

We report heterogeneity gains after training homogeneous and heterogeneous policies in Tab. 13. This shows that our curvature test predicts the heterogeneity gain of different reward structures, despite only being a component in the overall reward structure. This insight is important, as it may indicative our theoretical insights (the curvature test) may extend beyond environments that strictly follow our task allocation setting. However, this is just one possible scenario, and as such, this possibility requires further verification in future work.

The resulting policies are reported in Fig. 10 with videos here.

## N   OBSERVABILITY-HETEROGENEITY TRADE-OFF

In this Appendix, we crystallize the relationship between environment observability and empirical heterogeneity gains. It is well known that neurally homogeneous agents (i.e., sharing the same parameters) can achieve behavioral heterogeneity by conditioning their actions on diverse input contexts (behavioral typing). This can be achieved by naively appending the agent index to its observation (Gupta et al., 2017a) or by providing relevant observations that allows the agents to infer their role (Bettini et al., 2023). Behavioral typing is impossible in matrix games, as these games are observationless. However, it is possible in more complex games, such as our Multi-goal-capture scenario. We augment agents in the positive gain scenario ($U = \min, T = \max$) with a range sensor, providing proximity readings for other agents within a radius. In Fig. 11, we show that the

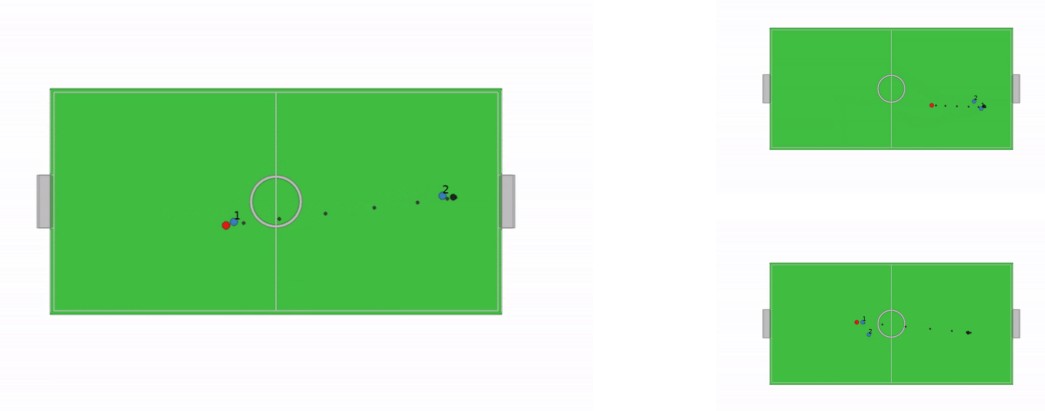

Figure 10: **Left:** Results of training a heterogeneous policy on VMAS Football where agents are trained with $U = \min, T = \max$ aggregators. Our learning agents are drawn in blue; the heuristic opponent in red; and the ball in black. The learning agents split their efforts, tackling both the ball and the opponent. **Right:** Results of training a homogeneous policy. Agents are unable to split their efforts, so either they both tackle the ball, or both tackle the opponent. This results in lower reward, hence $\Delta R > 0$. These policies are visualized on our website.

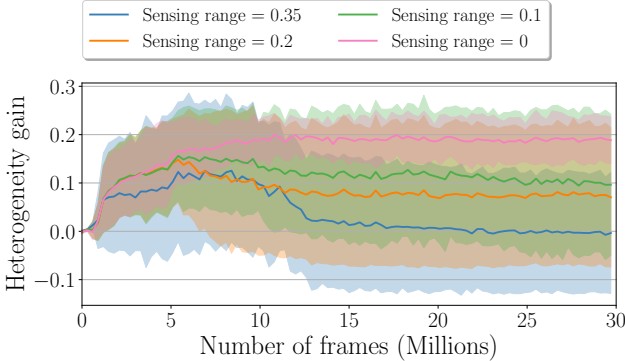

Figure 11: Gain w.r.t. observability when $U = \min, T = \max$.

Figure 12: Heterogeneity gain for Multi-goal-capture throughout training when the agents' observation range is gradually increased from 0 to 0.35 over 4 random seeds (4 random seeds suffice as this phenomenon is established in the literature (Bettini et al., 2023), and we only wish to show its emergence in the context of our work.)

heterogeneity gain decreases as the agent visibility increases (higher sensing radius). This is because, with a higher range, homogeneous agents can sense each other and coordinate to pursue different goals. This result highlights the tight interdependence between the heterogeneity gain and agents' observations.

## O PARAMETRIZED DEC-POMDP

A Parametrized Decentralized Partially Observable Markov Decision Process (PDec-POMDP) is defined as a tuple

$$\left\langle \mathcal{N}, \mathcal{S}, \{\mathcal{O}_i\}_{i \in \mathcal{N}}, \{\sigma_i^\theta\}_{i \in \mathcal{N}}, \{\mathcal{A}_i\}_{i \in \mathcal{N}}, \mathcal{R}^\theta, \mathcal{T}^\theta, \gamma, s_0^\theta \right\rangle_\theta,$$

where $\mathcal{N} = \{1, \ldots, n\}$ denotes the set of agents, $\mathcal{S}$ is the state space, and, $\{\mathcal{O}_i\}_{i \in \mathcal{N}}$ and $\{\mathcal{A}_i\}_{i \in \mathcal{N}}$ are the observation and action spaces, with $\mathcal{O}_i \subseteq \mathcal{S}, \ \forall i \in \mathcal{N}$. Further, $\{\sigma_i^\theta\}_{i \in \mathcal{N}}$ and $\mathcal{R}^\theta$ are the agent observation and reward functions, such that $\sigma_i^\theta : \mathcal{S} \mapsto \mathcal{O}_i$, and, $\mathcal{R}^\theta : \mathcal{S} \times \{\mathcal{A}_i\}_{i \in \mathcal{N}} \mapsto \mathbb{R}$. $\mathcal{T}^\theta$ is the stochastic state transition model, defined as $\mathcal{T}^\theta : \mathcal{S} \times \{\mathcal{A}_i\}_{i \in \mathcal{N}} \mapsto \Delta \mathcal{S}$, which outputs the probability $\mathcal{T}^\theta(s^t, \{a_i^t\}_{i \in \mathcal{N}}, s^{t+1})$ of transitioning to state $s^{t+1} \in \mathcal{S}$ given the current state $s^t \in \mathcal{S}$ and actions

$\{a_i^t\}_{i \in \mathcal{N}}$, with $a_i^t \in \mathcal{A}_i$. $\gamma$ is the discount factor. Finally, $s_0^\theta \in \mathcal{S}$ is a the initial environment state. A PDec-POMDP represents a set of traditional Dec-POMDPs (Oliehoek et al., 2016), where the observation function, the transition function, the reward function, and the initial state are conditioned on parameters $\theta$. This formalism is similar to the concepts of Underspecified POMDP (Dennis et al., 2020) and contextual MDP (Modi et al., 2018).

Agents are equipped with (possibly stochastic) policies $\pi_i(a_i|o_i)$, which compute an action given a local observation. Their objective is to maximize the discounted return:

$$G^\theta(\pi) = \mathbb{E}_\pi \left[ \sum_{t=0}^T \gamma^t \mathcal{R}^\theta \left( s^t, a^t \right) \middle| s^{t+1} \sim \mathcal{T}^\theta(s^t, a^t), a_i^t \sim \pi_i(o_i^t), o_i^t = \sigma_i^\theta(s_t) \right],$$

where $\pi, a$ are the vectors of all agents' policies and actions. $G^\theta(\pi)$ represents the expected sum of discounted rewards starting in state $s_0^\theta$ and following policy $\pi$ in a PDec-POMDP parametrized by $\theta$.

## P    STABILITY OF BILEVEL OPTIMIZATION IN HETGPS

The Heterogeneity Gain Parameter Search (HetGPS) algorithm employs a bilevel optimization framework to simultaneously optimize environment parameters and agent policies. This appendix discusses the structure of this optimization problem, its convergence properties, practical stability, and alternatives for non-differentiable environments.

### P.1    HETGPS AS A STACKELBERG GAME

HetGPS can be formalized as a Stackelberg game, a hierarchical optimization problem involving a leader and followers (Simaan & Cruz, 1973). In our setting:

1. The **Leader** is the environment designer (the outer loop of HetGPS), which aims to maximize the heterogeneity gain $HetGain^\theta$ by adjusting the environment parameters $\theta$.
2. The **Followers** are the homogeneous and heterogeneous multi-agent teams (the inner loop), which aim to maximize their respective returns $G^\theta(\pi)$ by optimizing their policies $\pi_{het}$ and $\pi_{hom}$ within the environment defined by $\theta$.

The leader's objective function (the heterogeneity gain) depends on the optimized policies of the followers, which, in turn, depend on the parameters $\theta$ set by the leader. Formally, the objective is:

$$\max_\theta \left[ G^\theta(\pi_{\text{het}}^*(\theta)) - G^\theta(\pi_{\text{hom}}^*(\theta)) \right] \tag{1}$$

where $\pi^*(\theta)$ represents the optimized policies for a given environment configuration $\theta$.

### P.2    CONVERGENCE AND STABILITY

Generally speaking, multi-agent reinforcement learning is a concurrent optimization process that faces non-stationarity as agents constantly adapt to one another's evolving policies (Zhang et al., 2019). HetGPS extends this challenge as agents must also adapt to a changing environment. Consequently, formal convergence guarantees to a **global** optimum remain an open question with regards to HetGPS in particular, but also MARL algorithms in general. However, recent theoretical work in environment co-design has established conditions under which convergence of bilevel optimization processes similar to HetGPS to **local** optima can be guaranteed, such as requiring sufficient smoothness of the environment dynamics and policy updates (Gao et al., 2024).

Despite the theoretical complexities inherent in multi-agent learning and bilevel optimization, as shown in Sec. 5 and App. Q, HetGPS demonstrates strong empirical stability even under adversarial initializations. This stability is expected, as it mirrors the practical success observed in related co-design and automated curriculum learning literature (Dennis et al., 2020; Gao et al., 2024). However, we emphasize that empirical stability is not a guarantee of convergence. Although we did not identify such cases ourselves, it is possible that in some scenarios, HetGPS will oscillate rather than converge.

### P.3    ADVANTAGE OF DIFFERENTIABLE SIMULATION

In our experiments, HetGPS increased training time by roughly 25% compared to training agents in an environment with a fixed reward structure. Hence, it is highly efficient and does not impose much

overhead. A key strength contributing to the efficiency of HetGPS is its use of differentiable simulation (e.g., VMAS (Bettini et al., 2022)). By leveraging backpropagation through the entire rollout, HetGPS computes the exact gradient $\nabla_\theta HetGain^\theta$. This approach is more sample-efficient than alternative methods that treat the environment design as a separate RL problem (e.g., PAIRED (Dennis et al., 2020) or Designer-RL (Gao et al., 2024; Amir et al., 2025)). Such methods rely on high-variance policy gradient estimates for the outer loop and often struggle with exploration inefficiency (Parker-Holder et al., 2021; Jiang et al., 2021; Xu et al., 2022). By utilizing exact gradients, HetGPS mitigates these issues.

### P.4  HANDLING NON-DIFFERENTIABLE ENVIRONMENTS

A requirement for the implementation of HetGPS presented in Alg. 1 is access to a differentiable simulator. When the environment involves non-smooth physics or black-box components, direct backpropagation is infeasible.

In such cases, the environment optimization step (Line 6 of Alg. 1) can be replaced with the gradient-free methods mentioned above, such as PAIRED (Dennis et al., 2020), the bilevel method from (Gao et al., 2024), or evolutionary strategies (Stanley et al., 2019). While these methods have empirically been shown to be stable and robust in other co-design settings, and may enable the extension of HetGPS to non-differentiable settings, they typically require more samples and may exhibit more noise compared to the direct backpropagation approach utilized in this work.

## Q  HETGPS UNDER ADVERSARIAL INITIAL CONDITIONS

To evaluate the robustness of HetGPS to initialization, we repeated the Softmax experiment in Multi-Goal-Capture (Fig. 5a) with adverse initialization. We initialized the outer aggregator $U$ with $\tau = 5$ (making it convex) and the inner aggregator $T$ with $\tau = -5$ (making it concave), which is the opposite of the concave-convex configuration predicted by theory to maximize heterogeneity gain.

As shown in Table 14, HetGPS successfully overcomes the adverse initialization and converges towards the theoretically optimal parameters (large positive $\tau$ for T, large negative $\tau$ for U).

Table 14: Convergence of HetGPS parameters ($\tau$) in the Softmax Multi-Goal-Capture experiment starting from adverse initialization ($\tau_T = -5, \tau_U = 5$). Mean and standard deviation reported over 3 seeds.

| Frames (M) | 0 | 50 | 75 | 100 |
|---|---|---|---|---|
| $\tau$ of T (Inner Agg.) | $-5.0 \pm 0.0$ | $13.32 \pm 2.15$ | $18.88 \pm 3.96$ | $22.95 \pm 5.71$ |
| $\tau$ of U (Outer Agg.) | $5.0 \pm 0.0$ | $-10.26 \pm 1.70$ | $-14.59 \pm 2.24$ | $-17.16 \pm 2.43$ |

## R  BEST PRACTICES, LIMITATIONS, AND OPEN QUESTIONS

We list a number of limitations, open questions, and possible extensions. We then discuss best practices: where and how should our theoretical results be applied? What parameters should HetGPS optimize?

### R.1  THEORETICAL SCOPE

- **Beyond task-allocation RL domains.** The benchmark domains we study and the additional settings covered in App. E all fit into our abstract task-allocation framework: we can interpret agents' state, such as goal proximity in Multi-goal-capture, or whether they captured an escaping agent in tag, abstractly as "efforts" $r_{ij}$ and represent the reward in terms of such efforts. This is what enables us to make predictions about these environments. Although our framework is quite general, and accommodates environments that one might not traditionally view as "task allocation" (such as football and tag), several notable multi-agent RL domains, e.g., multi-robot manipulation, might not be representable within this framework. Our heterogeneity analysis does not directly apply to these settings, and extending our results to them is important for getting a complete picture of the benefits of heterogeneity.

## R.2 ALGORITHMIC ASSUMPTIONS

- **Differentiable simulation.** HetGPS requires $\nabla_\theta G^\theta(\pi)$, hence a simulator that is end-to-end differentiable and tractable to back-propagate through. We assume differentiability mainly for considerations of training efficiency. However, many realistic environments still rely on non-smooth physics or black-box generators, requiring us to modify HetGPS for these settings. We note that there are good, established methods for learning environment parameters in non-differentiable settings (at the cost of efficiency/increased noise). We discuss these in detail in App. P. However, we did not test such variants of HetGPS, and leave these extensions to future work.

## R.3 OPEN QUESTIONS

i. **What is the connection between the transition function and heterogeneity?** Our analysis is reward-centric: the curvature criterion reasons only about the team reward. In a Dec-POMDP, however, heterogeneity can be beneficial purely because agents are constrained by *state transitions*. When do state transition dynamics benefit heterogeneity?

ii. **Learning dynamics vs. reward structure.** The theory predicts whether a given reward structure *enables* an advantage to heterogeneous teams, not whether a particular learning algorithm will learn in response to it. This is connected to the difference between *neural* and *behavioral* heterogeneity that we emphasize throughout the paper. Our experiments suggest, empirically, that neurally heterogeneous agents will, in practice, learn to exploit heterogeneous reward structures (i.e., be behaviorally heterogeneous); but can a formal link be established between our reward structure insights and what reward the learning dynamics converge to in practice?

Tackling these challenges would sharpen our understanding of *when* and *how* diversity should be engineered in cooperative multi-agent learning.

## R.4 SCOPE OF THE CURVATURE ANALYSIS

The theoretical framework presented in Section 2 provides a precise characterization of the heterogeneity gain based on the curvature of reward aggregators. We provide an extended discussion of when we expect our theoretical predictions can, and cannot, be applied for deciding whether to use heterogeneous or homogeneous agent policies.

**Symmetry and Monotonicity.** Our analysis hinges on the definition of generalized aggregators as symmetric and coordinate-wise non-decreasing. These assumptions are appropriate for studying emergent behavioral specialization among capability-identical agents. Symmetry ensures that agents (and tasks) are interchangeable *ex-ante*, isolating how the reward structure drives specialization. If symmetry is violated (e.g., due to inherently heterogeneous agent capabilities), heterogeneity is often trivially necessary. Monotonicity ensures a rational cooperative setting where increased effort does not decrease the reward.

**Effort Constraints.** We define the feasible effort space over the closed unit simplex (where efforts sum $\leq 1$). However, because both the inner aggregators $T_j$ and the outer aggregator $U$ are non-decreasing, any optimal allocation—whether homogeneous or heterogeneous—will necessarily saturate the budget constraint (efforts sum $= 1$). Therefore, our analysis focuses on this efficient frontier without loss of generality.

**Constant-Sum Task Score Constraints.** It is crucial to clarify that the assumption of constant-sum task scores ($\sum_j T_j(a_j) = C$) is specific only to Theorem 3.3, enabling the use of majorization to analyze the outer aggregator $U$. Theorems 3.1 and 3.2, and our sum-form aggregator analysis (App. F), do not rely on this.

When this assumption is violated, the analysis of $\Delta R$ involves a trade-off between the distribution of scores (influenced by curvature) and their total magnitude. Despite this complexity, our empirical findings (Section 5) suggest that the curvature analysis remains a robust heuristic for predicting the heterogeneity gain even when the task scores are variable-sum.

**Reward functions that do not precisely follow the theory.** Our football experiments show that even when only part of the reward function adheres to our curvature theory (e.g., it is a sum $R(A) = R_1(A) + R_2(A)$ where $R_1$ is concave-convex and $R_2$ is a function with unclear curvature), our theoretical results may still predict the heterogeneity gain. We make no formal claims about the robustness of our predictions in such scenarios, but it is valuable to keep in mind that even if the entire reward function does not perfectly follow the theory, it may still be worthwhile to see what the concave-convex curvature test says about the part of it that does.

