# OpenReview forum: "When Is Diversity Rewarded in Cooperative Multi-Agent Learning?"
_ICLR.cc/2026/Conference — ICLR 2026 Poster_

### Official Review · Reviewer_icAF · 2025-10-24

**Soundness:** 3
**Presentation:** 3
**Contribution:** 3
**Rating:** 6
**Confidence:** 4

**Summary:**

This paper presents a principled framework for understanding when and why behavioral diversity among cooperative agents leads to improved collective performance. The authors introduce the notion of heterogeneity gain, defined as the performance difference between heterogeneous and homogeneous teams, and formally analyze it within a general two-level reward aggregation model. Using majorization theory and Schur-convexity analysis, they prove that the curvature of the inner and outer aggregation functions determines whether diversity enhances or hinders performance — specifically, concave-outer and convex-inner structures favor heterogeneity. Building on this theoretical foundation, the paper proposes HetGPS (Heterogeneity Gain Parameter Search), a differentiable environment co-design algorithm that discovers reward parameterizations maximizing or minimizing heterogeneity gain through bilevel optimization. Empirical studies across analytic matrix games and embodied MARL tasks (multi-goal capture, tag, and football) validate the theory, showing that HetGPS consistently rediscovers curvature conditions predicted to reward diverse behaviors. The work provides a unifying explanation for when diversity benefits cooperative learning and establishes a new paradigm linking reward curvature, environment design, and emergent specialization in multi-agent systems.

**Strengths:**

1. The paper introduces a mathematically precise metric $\Delta R = R_{\text{het}} - R_{\text{hom}}$​ to quantify when heterogeneous (specialized) teams outperform homogeneous (identical) teams. It proves that the sign of ΔR depends solely on the Schur-convexity or concavity of the inner and outer reward aggregators: (1) Schur-convex inner operator leads to heterogeneity gain; (2) Schur-concave inner operator leads to no gain; (3) Schur-convex outer operator (under constant-sum normalization) leads to no gain. This gives a simple convexity test for when diversity helps — a clear theoretical advance over heuristic assumptions in prior MARL work.
2. This paper provides closed-form and bounded results for Softmax, Power-Sum, and {min, mean, max} families, creating a general mathematical map of reward structures that promote or suppress diversity.
3. A new gradient-based environment co-design algorithm that directly optimizes environment parameters to maximize or minimize $\Delta R$: (1) It operates via differentiable simulators (back-propagates through environment); (2) It avoids inefficiencies and instability of RL-based environment design (e.g., PAIRED); (3) It can either promote or suppress heterogeneity by switching gradient ascent/descent. The algorithm alternates MARL policy training and environment updates efficiently — roughly 25 % overhead — and robustly rediscovering the theoretical curvature configurations.
4. Tests across (i) analytical matrix games, (ii) embodied long-horizon MARL tasks (Multi-Goal Capture, 2v2 Tag, Football). All experiments confirm theoretical predictions: **concave-outer + convex-inner** reward structures yield the largest heterogeneity gains. The algorithm autonomously learns the same curvature conditions predicted by theory (inner convex, outer concave) for both Softmax and Power-Sum parameterizations — a strong consistency check.
5. This paper links convex analysis and majorization theory to concrete MARL reward design, turning diversity from a heuristic into a controllable design variable. Also, it provides a principled answer to an open question in cooperative MARL — under what reward structures does behavioral diversity outperform homogeneity?
6. The contribution of this paper is applicable beyond RL to any cooperative optimization setting expressible as nested aggregations (e.g., task allocation, team composition, distributed robotics).
7. Complete code and YAML configurations provided; all mathematical proofs and assumptions detailed in appendices. Figures (e.g., Fig. 2–5) clearly demonstrate the alignment between theoretical curvature analysis and empirical $\Delta R$ outcomes. Writing is logically structured, connecting intuition, theory, and experiment in a coherent flow.

**Weaknesses:**

1. While elegant, the use of Schur-convexity assumes symmetric and differentiable reward functions. Many practical rewards in MARL are asymmetric, piecewise, or sparse — outside the smooth function classes covered by majorization theory.
2. Though claimed to be “25% higher,” the method still requires dual policy training (heterogeneous vs homogeneous teams) and outer-loop environment gradient updates. The bilevel optimization may become prohibitively expensive for large-scale, high-dimensional MARL tasks.
3. Empirical validation is confined to simple matrix games and small-scale MARL benchmarks (Multi-goal Capture, 2v2 Tag, Football mini-games). There are no large-scale or high-dimensional tasks (e.g., SMACv2, MPE with >10 agents, or real robotic control) to demonstrate robustness.

**Questions:**

> Please answer the following questions:

1. The theory presumes identical agents differing only in behavior (i.e., “behavioral heterogeneity”). What if agents differ in capacity, observation space, or action bounds? Would Schur-convexity still capture heterogeneity gain?
2. The method performs bilevel optimization: inner MARL loop + outer gradient update of environment parameters. Does the paper analyze or guarantee convergence, or could gradient coupling cause oscillations?
3. HetGPS successfully rediscovers concave-outer/convex-inner curvature. Would it also succeed if reward functions were implemented via neural networks instead of analytic forms?

> The possible suggestions:

A prior work [1] also found that the reward structure (even encoding the same goal) in a real-world problem could impact the performance of MARL algorithms a lot, though it does not consider the multi-task scenarios. This can be used as an example to motivate the research goal of this paper.

[1] Wang, Jianhong, Wangkun Xu, Yunjie Gu, Wenbin Song, and Tim C. Green. "Multi-agent reinforcement learning for active voltage control on power distribution networks." _Advances in neural information processing systems_ 34 (2021): 3271-3284.

The global reward structure that aggregates reward functions embedding different subtasks/roles is highly relevant to payoff allocations/credit assignment. Especially, in prior work MARL algorithms with Shapley values [2] also demonstrate that agents with heterogenous behaviors is corresponding to different payoffs assigned by Shapley values. This paper aims to automatically search for a design of local rewards which is an reverse process against the prior work, that design local rewards inspired by cooperative game theory leading to heterogeneous behaviors. Thus, this relevant prior work is encouraged to be discussed in related work.

[2] Wang, Jianhong, Yuan Zhang, Tae-Kyun Kim, and Yunjie Gu. "Shapley Q-value: A local reward approach to solve global reward games." In _Proceedings of the AAAI conference on artificial intelligence_, vol. 34, no. 05, pp. 7285-7292. 2020.

---

> ### Author Response · Authors · 2025-11-21
> **Response to Review icAF, Part 1/2**
>
> We thank the reviewer for the detailed feedback and helpful suggestions. We updated the revised manuscript in a number of parts based on this feedback.
>
> **W1: While elegant, the use of Schur-convexity assumes symmetric and differentiable reward functions. Many practical rewards in MARL are asymmetric, piecewise, or sparse.**
>
> *   **Symmetry:** Regarding asymmetry, this often makes the problem easier rather than harder. If agents have different capabilities or goals, heterogeneity is often trivially necessary. We focus on the more challenging case of capability- and goal-identical agents (Sec. 2), where symmetry allows us to isolate the impact of the reward structure itself.
>
> *   **Differentiability/Piecewise:** While some tests for Schur-convexity rely on derivatives, the concept of majorization itself does not require differentiability. Our analysis covers non-differentiable aggregators like `min` and `max` (Fig. 2 analysis, App. H).
>
> *  **Sparse rewards:** Our experiments directly demonstrate that the curvature insights hold in environments with sparse and piecewise rewards, both in the embodied long-horizon setting (Tag in Sec. 5.(ii-2)) and the unembodied instantenous setting (discrete matrix games in Sec 5.(i)).
>
> **W2: The bilevel optimization [HetGPS] may become prohibitively expensive for large-scale, high-dimensional MARL tasks.**
>
> We believe it is important to distinguish between HetGPS (the algorithm) and the core theoretical insights.
>
> *   **The Theory Scales:** Our aggregator theory (Sec. 3) scales to any number of agents (N) and any number of tasks (M) and is the key analytical tool we introduce.
> *   **HetGPS as a Design Tool:** HetGPS is used here as a tool for *discovering* reward functions. It is acceptable if HetGPS itself does not scale to very large dimensionalities: we can run it on a lower-dimensional case (e.g., fewer agents) to gain insights, and then use the reward function it designs in a higher-dimensional case.
>
> Regarding computational scaling:
> *   **Number of agents (N):** When training heterogeneous agents, the parameter count scales linearly with N (as in any MARL paradigm training independent policies). This is intrinsic to MARL training.
> *   **High-dimensionality (Observation/Action Spaces):** The growth of these spaces affects our method no differently than any standard MARL algorithm.
>
> In summary, we do not see a reason why our approach should be affected by scale and dimensionality more severely than standard MARL practices.
>
> **W3: Empirical validation is confined to small-scale MARL benchmarks... no large-scale or high-dimensional tasks (e.g., >10 agents, or real robotic control).**
>
> *   **Larger Scale:** The revised version of the paper now contains two experiments with **11 agents**: 8v3 Tag (App. L), and discrete matrix games with N=11 (App. J). Both scenarios closely follow our theoretical predictions.
> *   **Environment Complexity (Robotics):** The Multi-goal-capture, Football, and Tag scenarios are continuous control robotics tasks from the VMAS simulator. It has been shown that policies trained on VMAS can be deployed in a zero-shot fashion to real-world DJI Robomaster robots [1,2]. The continuous control nature of these tasks highlights their complexity and relevance to real robotic applications, arguably more so than video games like SMAC which often use discrete actions.
> *   **Real Robot Deployment:** As we are not proposing a new sim-to-real pipeline, basic deployment of VMAS policies on real robots has already been demonstrated [1,2].
>
> [1] Blumenkamp, et al. "The Cambridge RoboMaster". DARS 2025.
>
> [2] Bettini, et al. "Heterogeneous Multi-Robot Reinforcement Learning". AAMAS 2023.
>
> **Q1: The theory presumes identical agents... What if agents differ in capacity, observation space, or action bounds?**
>
> This relates to the discussion on asymmetry (W1). If agents differ in capacity (e.g., Agent A is inherently better at Task 1), heterogeneity is often trivially optimal, regardless of the reward curvature. The contribution of our paper is analyzing the harder problem: when do *identical* agents benefit from specializing? Schur-convexity analysis specifically isolates how the reward structure incentivizes specialization independent of inherent capability differences.
>
> **Q2: Oscillations in bilevel optimization**
>
> The convergence and stability of the bilevel algorithm are discussed in **App. P.2**. To summarize: since we cannot guarantee convergence of MARL training itself (the follower problem), it is impossible to guarantee convergence of the overall game.
>
> However, we demonstrate strong empirical stability, and have not observed coupling-caused oscillations empirically. Furthermore, in **App. Q**, we show that HetGPS robustly converges even when reward aggregators are initialized in an adverse configuration. We agree that, in theory, oscillation might emerge in certain cases, and we have further emphasized this in the revised Appendix P.

---

> > ### Author Response · Authors · 2025-11-21
> > **Response to Review icAF, Part 2/2**
> >
> > **Q3: Would [HetGPS] also succeed if reward functions were implemented via neural networks instead of analytic forms?**
> >
> > Thank you for the great question. It depends on what is intended by "success." HetGPS can optimize any differentiable environment parametrization, including unconstrained neural network reward functions. In such a case, the optimization space is so vast that HetGPS could likely achieve increasingly larger heterogeneity gain (potentially towards infinity).
> >
> > However, the reward functions obtained this way would likely not be interpretable. The goal of this work is to understand *why* heterogeneity is rewarded. Constraining the reward functions to the parametrized families studied in the theory allows us to interpret their meaning, while letting HetGPS optimize the parameter that the theory highlights as crucial: their curvature.
> >
> > **Suggestions: Related Work [1] and [2].**
> >
> > Thank you for pointing out these excellent works, which provide valuable context for our research.
> >
> > Wang et al. (2021) [1] illustrates how reward structure impacts MARL performance. Wang et al. (2020) [2] provide a game-theoretic approach (Shapley values) to decomposing global rewards, demonstrating that fair credit assignment often corresponds to heterogeneous behaviors.
> >
> > Our work is complementary: while [2] focuses on how to efficiently decompose a given reward value, we analyze the properties of the global aggregation function itself (its curvature) to predict when such heterogeneous specialization is inherently beneficial.
> >
> > We have updated the **Related Work section (Sec 1.1)** to discuss these papers.
> >
> > We welcome any further questions or suggestions for improving our manuscript, and would like to thank the reviewer again for the insightful review and references.

---

> ### Comment · Reviewer_icAF · 2025-11-22
>
> I appreciate your detailed reponse. Most of my concerns and questions have been addressed, however, there are still some remaining doubts about part of your answers:
>
> **1. HetGPS as a Design Tool: HetGPS is used here as a tool for discovering reward functions. It is acceptable if HetGPS itself does not scale to very large dimensionalities: we can run it on a lower-dimensional case (e.g., fewer agents) to gain insights, and then use the reward function it designs in a higher-dimensional case.**
>
> I agree that this could be a pragmatic plan, but what I am curious about is: how do you promise that the local charateristics can still be guaranteed in the global scope?
>
> **2. In summary, we do not see a reason why our approach should be affected by scale and dimensionality more severely than standard MARL practices.**
>
> I agree with your words, but the main extra computational overhead (inference) may led by the rollout stage, as you introduce **another team of homogeneous agents**. This may lead to double efforts on sample collection from the environment, and this would become worse when the scale of agents becomes larger. This is what the common MARL pipeline does not possess.

---

> ### Author Response · Authors · 2025-11-23
>
> We thank the reviewer for these insightful follow-up questions regarding the efficacy and overhead of HetGPS.
>
> **1. Transferability of HetGPS's discovered features across scales**
>
> Great question. HetGPS can choose which parameters of the PDec-POMDP it optimizes. To enable transfer from low-dimensional cases to higher-dimensional ones, we must choose parameters that are scale-invariant to the number of agents and tasks.
>
> Concretely, in this work, we use HetGPS to optimize the parameters controlling the curvature (e.g., $\tau$) of the aggregators $U$ and $T$. Our theoretical results (Theorems 3.1-3.3) formally guarantee that the relationship between this curvature and the heterogeneity gain (ΔR>0) is scale-invariant. Therefore, when HetGPS discovers an optimal curvature configuration for a small number of agents and tasks, the theory ensures this finding is valid for any number.
>
> More broadly, HetGPS is an empirical tool for discovering environment features (reward or non-reward related) that result in ΔR>0. For its findings to be generalizable, we advise practitioners to focus on parameters known or hypothesized to be scale invariant. Additional, non-curvature-related parameters that arose in our work which we suspect to have a scale-invariant impact on whether ΔR>0 are **(i)** agents’ sensing range, studied in App. N, and **(ii)** possibly the coefficient ‘β’, balancing progress vs. specialization in the reward function for Football in App. M. While the relationship between these non-curvature-related parameters and heterogeneity is not our focus in this paper, we view them as exciting targets for study using HetGPS in future work.
>
>
> **2. Computational overhead of HetGPS**
>
> The reviewer is correct that HetGPS requires training two teams, doubling the environment interactions (rollouts) compared to a single MARL training run. However, the actual computational impact is significantly mitigated by several factors:
>
> * **Homogeneous training is efficient:** The homogeneous team utilizes full parameter sharing (training only $1$ network instead of $N$). This makes the homogeneous team’s training scalable and minimizes the marginal cost as $N$ increases. In effect, rather than training $2N$ networks we are training $N+1$ networks ($N$ for the heterogeneous team and $1$ for the homogeneous team), reducing training duration significantly.
>
> * **Complexity scaling:** Importantly, we are training two independent $N$-agent teams in parallel environments, *not* a single $2N$-agent team interacting in the same environment. This enables us to take advantage of parallelization and vectorization (see below).
>
> * **Vectorization of simulator:** We leverage vectorized simulation (i.e., VMAS) and vectorized neural network inference. This means the physics engine operates over tensors of shape [M,N,...], where M is the number of environments and N is the number of agents. Therefore, simulating more parallel environments or agents in VMAS only slightly increases the computational cost. This is because the cost of a physics step is invariant wrt M and N (as long as the tensors fit in GPU memory) thanks to the SIMD paradigm of GPU computation.
>
> These factors are why, as noted in the paper, we only witnessed a modest (25\%) increase in runtime in our experiments deploying HetGPS vs. training a heterogeneous team on a fixed environment.
>
> We appreciate the discussion about these nuanced points and welcome further questions or comments. We will add the above  clarifications in the paper.

---

> > ### Comment · Reviewer_icAF · 2025-11-23
> >
> > I thank the authors for the well-described answers. All these engineering techniques to remove the potential limitation of the proposed HetGPS are sensible.

---

### Official Review · Reviewer_M2TT · 2025-10-30

**Soundness:** 3
**Presentation:** 3
**Contribution:** 3
**Rating:** 6
**Confidence:** 3

**Summary:**

This paper addresses the critical and previously unresolved question of when behavioral heterogeneity is inherently beneficial in cooperative multi-agent task allocation problems. The authors provide a principled theoretical answer by demonstrating that the potential advantage of specialized policies is determined by the curvature of the global reward function, which is modeled as a double aggregation of agent efforts. Specifically, positive heterogeneity gain is predicted when the inner, task-level aggregator is Schur-convex and the outer, team-level aggregator is Schur-concave. To validate and extend these insights to complex, time-extended environments, the paper introduces HetGPS (Heterogeneity Gain Parameter Search), a gradient-based bilevel optimization algorithm. HetGPS efficiently searches the parameter space of rewards and successfully rediscovers the theoretically optimal reward regimes in various MARL settings, thereby connecting the abstract theoretical predictions directly to reward design in practical environments5. The results ultimately offer clear criteria for diagnosing and designing missions where agent diversity is essential for maximizing team performance.

**Strengths:**

1. This paper is well-written and easy to follow. The author provide sufficient supplementary material, making the conclusions of this paper clearer and more convincing.
2. This paper provides a rigorous, formal theory for predicting when behavioral heterogeneity is advantageous in multi-agent task allocation problems. This theoretical framework, based on the curvature of reward aggregation operators (Schur-convexity/concavity), moves the selection of diversity from ad-hoc heuristics to a principled design dimension.
3. The introduction of Heterogeneity Gain Parameter Search (HetGPS) is a significant algorithmic contribution. This gradient-based bilevel optimization method efficiently searches the reward parameter space to find configurations that maximize or minimize the empirical heterogeneity gain.
4. The extensive experiments, ranging from abstract matrix games to complex embodied MARL scenarios (Multi-goal-capture, Tag, Football), successfully demonstrate that the theoretical predictions derived from reward curvature reliably transfer to long-horizon settings. HetGPS further validates the theory by automatically discovering the predicted optimal reward regimes.

**Weaknesses:**

1. The core theoretical criterion for heterogeneity gain is based solely on the curvature of the reward function (Schur-convexity/concavity). This analysis is inherently restricted to the reward structure and does not formally integrate the complexity of environment dynamics.
2. The high efficiency and tractability of the HetGPS algorithm fundamentally rely on the assumption of an end-to-end differentiable simulator. This is required to compute the exact environment gradients via backpropagation.

**Questions:**

1. The two issues mentioned in the Weaknesses section.
2. The curves in Figure 4 overlap significantly, making them difficult to distinguish clearly.

---

> ### Author Response · Authors · 2025-11-21
>
> We thank the reviewer for the helpful questions and comments.
>
> **W1: The core theoretical criterion... is inherently restricted to the reward structure and does not formally integrate the complexity of environment dynamics.**
>
> We agree that our theoretical criteria are focused on the reward function, which we believe is highly important in itself. A key finding of our work is that the reward function alone is highly predictive of the advantage of heterogeneity, even in the presence of complex dynamics. Our empirical results consider a variety of scenarios with different, complex environment dynamics: Football, Tag, Multi-goal navigation, and matrix games. In *all* these scenarios, our theoretical criteria regarding reward curvature match the empirical heterogeneity gain.
>
> Formulating a predictive theory that couples the influence of both environment dynamics and the reward function is very challenging, and this is a direction we are interested in exploring in future work (discussed in App. R).
>
> **W2: The high efficiency and tractability of the HetGPS algorithm fundamentally rely on the assumption of an end-to-end differentiable simulator.**
>
> Yes, that is correct. We leverage differentiable simulation for efficiency, allowing HetGPS to compute exact gradients via backpropagation. We have tried to comprehensively address the limitations and alternative approaches in **Appendix P**. App. P.3 discusses the efficiency advantages gained via differentiability, and App. P.4 discusses alternative approaches (such as Evolutionary Strategies or treating environment design as an RL problem) when a differentiable simulator is not available. While these alternatives are less efficient, they enable the extension of the HetGPS concept to non-differentiable settings.
>
> **Q2: The curves in Figure 4 overlap significantly, making them difficult to distinguish clearly.**
>
> To improve clarity in this Figure, we have added tables (**Tabs.10, 11**) reporting the final gain values for each of the plots shown. These tables are referenced in the figures’ captions.
>
> We welcome any further questions or suggestions for improving our manuscript.

---

### Official Review · Reviewer_j8eg · 2025-11-01

**Soundness:** 3
**Presentation:** 2
**Contribution:** 3
**Rating:** 6
**Confidence:** 3

**Summary:**

This paper provides a theoretical analysis of the heterogeneous vs homogeneous credit assignment in MARL, which provides insights for the reward shaping problem in different MAS tasks. It presents HetGPS to optimize the parameter space of an underspecified MARL environment. The experiments showcase that it can discover new reward regimes to maximize the advantage of heterogeneity.

**Strengths:**

It is novel to formulate diverse reward allocation choices to a mathematical curvature question via Schur-convex/concave tools. The theorems/constructive counter-examples are clean with explicit assumptions. The algorithm description and experiment analysis are clearly presented. This work has significance in influencing environment/reward design and architecture choices in MARL.

**Weaknesses:**

1. Results hinge on symmetry/coordinate-wise monotonicity and near constant-sum task scores. It would be good to tabulate common benchmarks that violate these assumptions and provide bounds or heuristics for the reward difference when constant-sum fails.
2. Longer-horizon Dec-POMDP dynamics may interact with curvature in nontrivial ways; more systematic ablations or counterexamples would strengthen the claim
3. Figures all consist of 9 cases, making it difficult to distinguish the lines

**Questions:**

1. What are the conservative bounds $\Delta R$ when constant-sum is violated?
2. Will there be any cases where curvature is only a part of the total reward yet remains predictive?
3. In non-differentiable simulators, how do sample complexity and final performance compare when using score-function/ES estimators?
4. For non-symmetric or non-monotone aggregators (e.g., capacity/bottleneck constraints), will the results admit first-order approximations or sharp counter-examples?

---

> ### Author Response · Authors · 2025-11-21
> **Response to Review j8eg, Part 1**
>
> We thank the reviewer for the helpful questions and constructive comments.
>
> **W1: Results hinge on symmetry/coordinate-wise monotonicity and near constant-sum task scores. It would be good to tabulate common benchmarks that violate these assumptions.**
>
> Thank you for this excellent suggestion. In the revised paper, we have updated **Appendix R** with an extended discussion of when our assumptions apply. To summarize:
>
> *   **Symmetry:** Our work studies when *capability-identical* agents with the same goal benefit from heterogeneity. Symmetry ensures that agents (and tasks) are interchangeable, isolating how the reward structure drives specialization. If asymmetry is present (e.g., inherently heterogeneous capabilities), heterogeneity is often trivially necessary, meaning analytical tools like ours may not be needed to identify the benefit.
> *   **Monotonicity:** This ensures a rational cooperative setting where increased effort does not decrease the reward. This assumption is broadly applicable to cooperative multi-agent scenarios.
> *   **Task Score Constraints:** It is important to clarify that the assumption of constant-sum task scores ($\sum_j T_j(a_j) = C$) is **specific only to Theorem 3.3**. Our main results, Theorems 3.1 and 3.2, do *not* rely on this. Many scenarios violate this assumption, including our embodied experiments in Section 5 (Tag, Football, Multi-goal-capture).
>
> **W2: Longer-horizon Dec-POMDP dynamics may interact with curvature in nontrivial ways.**
>
> We agree that the interaction between dynamics and reward curvature is complex and an important open question (discussed in Appendix R.3.i). Our analysis focuses primarily on the reward structure. However, our empirical results across diverse, long-horizon environments (Multi-goal capture, Tag, Football) demonstrate that the reward curvature remains highly predictive even when complex dynamics are present. Additionally, in the revised paper, the Tag experiments (**Appendix L**) now include an 8v3 scenario with episodes twice as long as the original experiments (1000 steps), confirming the stability of our predictions over extended horizons. Developing a unified theory coupling dynamics and reward curvature is a challenging direction for future work.
>
> **W3: Figures all consist of 9 cases, making it difficult to distinguish the lines.**
>
> Thank you for pointing this out. In the revised paper, tables are referenced in the figure captions and report the final gain values for each of the plots shown.
>
> **Q1: What are the conservative bounds when constant-sum is violated?**
>
> As noted in W1, all our results except Theorem 3.3 do not assume task scores are constant-sum (efforts are also not required to be constant-sum, see the response to reviewer JKoM). Hence, non-constant-sum task scores typically do not affect the applicability of our main analytical results (Thms 3.1 and 3.2).
>
> More generally, regarding robustness when assumptions are violated, our Football experiments (App. M) show that even when the reward function only partially adheres to our theory (e.g., $R(A)=R_1(A)+R_2(A)$ where $R_1$ fits the curvature analysis and $R_2$ does not), our theoretical results may still predict the heterogeneity gain.
>
> **Q2: Will there be any cases where curvature is only a part of the total reward yet remains predictive?**
>
> Yes, precisely. Our Football experiments in **App. M** are a key example of this. The reward is composed of agent-wise components fitting the curvature formulation and a global component that does not follow our theory. Even in this mixture, the curvature test remains highly predictive of the outcome, highlighting the robustness of our insights.
>
> **Q3: In non-differentiable simulators, how do sample complexity and final performance compare when using score-function/ES estimators?**
>
> This is discussed in **Appendix P.4**. While we have not explicitly studied such simulators, based on the literature on gradient-free optimization and co-design, we anticipate that using these methods (e.g., PAIRED or evolutionary strategies) would be stable and produce similar final results to HetGPS. However, the training process would most likely be noisier and significantly less sample efficient compared to the direct backpropagation approach utilized in this work.

---

> ### Author Response · Authors · 2025-11-21
> **Response to Review j8eg, Part 2/2**
>
> **Q4: For non-symmetric or non-monotone aggregators (e.g., capacity/bottleneck constraints), will the results admit first-order approximations or sharp counter-examples?**
>
> *   **Asymmetry:** As noted in W1, in scenarios with asymmetry, such as per-agent capacity constraints, heterogeneity is likely to be trivially beneficial. We study the more challenging problem of identifying when heterogeneity is necessary assuming *identical* agent capabilities and goals.
> *   **Monotonicity:** Our monotonicity constraint states that increasing effort in a task should not decrease the task-level score. This assumption seems widely applicable in cooperative settings. We are unable to think of natural scenarios violating this constraint and would be happy to discuss specific examples if the reviewer has them in mind.
>
> We welcome any further questions or suggestions for improving our manuscript.

---

### Official Review · Reviewer_JKoM · 2025-11-02

**Soundness:** 4
**Presentation:** 4
**Contribution:** 3
**Rating:** 10
**Confidence:** 3

**Summary:**

This paper investigates aggregation functions in a multi-task/multi-objective multi-agent RL setting, particularly answering the question of for which aggregations heterogeneous behavior is advantageous over homogeneous behavior.
A theoretical analysis shows that this is connected to the schur-convexity of the aggregation functions.
Experiments confirm the theoretical analysis.
Further, a method to optimize environment parameters such that they favor heterogenity is proposed and validated.

**Strengths:**

* The problem setting is interesting and relevant. The question of whether or not to share parameters in Multi-Agent RL is relevant and the analysis of aggregation functions in this paper provides a useful step towards answering it.
 * The theoretical analysis and its presentation are very clear and well explained
 * The experiments nicely confirm the theoretical predictions.

**Weaknesses:**

Minor points:
 * The assumption of normalized inner aggregators could be justified better. It's not entirely clear to me whether this is justified in practice

**Questions:**

*  See question about assumption in weaknesses
 * In HetGPS, it is not entirely clear to me why an approach that alternates between policy and environment improvement was chosen. Intuitively, it could be posed as an entirely bi-level process, in which the policies are trained from scratch for each environment configuration


Nitpicks:
 * L71 It may be better to introduce the symbols T and U for agent-wise and team-wise aggregation here already, on a first read the two addition symbols are a bit unclear.
 * L172 Similar issue, the abuse of notation is initially confusing and (in my opinion) unnecessary
 * L177 Introducing the allocations under the close simplex, i.e. allowing for sum < 1, and then excluding this case in L207 seems unnecessary
 * L351 "continues to be and informative"

Overall I really enjoyed reading this paper, great work!

---

> ### Author Response · Authors · 2025-11-21
>
> We thank the reviewer for the helpful questions and suggestions for improvement. We are glad you enjoyed our paper!
>
> **Q1/W1: Is the assumption of normalized inner aggregators justified in practice?**
>
> We assume this refers to **Theorem 3.3**, as we do not assume normalized (constant-sum) inner aggregators outside this specific theorem. Please let us know if we misunderstood the question.
>
> Inner aggregators are normalized in several notable cases. For example: (i) $T_i$ is the sum operation $\sum$ (since total effort is constant), or (ii) the task scores $(T_1, \ldots, T_M)$ represent a probability distribution (as in our Softmax analysis, Thm 3.4).
>
> In general, we agree that Theorem 3.3 is less broadly applicable than our other core results (like Thms 3.1, 3.2, or the results for sum-form operators in Appendix F), which do not require this normalization assumption and cover scenarios like our embodied experiments in Section 5.
>
> **Q2: In HetGPS, why was an approach that alternates between policy and environment improvement chosen, rather than training policies from scratch for each environment configuration?**
>
> We agree that ideally, retraining policies from scratch for every new environment configuration would be the most robust approach. However, this makes the algorithm computationally far more expensive. Since environment updates are gradient-based, they perturb parameters locally. We thus assume policies from the previous configuration provide a good starting point (warm-start) for the next iteration. However, if users wish to use the more expensive version of HetGPS, our code already supports resetting agents’ policies every environment iteration; it is sufficient to enable the `cfg.model.reset_with_env` parameter.
>
> **“Nitpicks” (Notation and Closed Simplex):**
>
> Thanks for pointing these out!
>
> *   **Notation (L71/L172):** We have taken your advice and now prioritize the $U$ and $T$ notations in most places in our paper, minimizing the use of the $\bigoplus$ notation to improve clarity.
> *   **L177 (Closed Simplex):** We consider the entire closed simplex ($\sum r_{ij} \le 1$) because in practice, agents may realize effort allocations that do not sum to 1. For example, in 2v2 Tag, if no chaser agent manages to capture an escaper, the realized efforts sum to 0. While such allocations are suboptimal from a theoretical optimization viewpoint (as noted in L207), they occur during the learning process and in practice, and our model accommodates this.
>
> We welcome any further questions or suggestions for improving our manuscript.

---

### Author Response · Authors · 2025-11-21

We thank the reviewers for their insightful comments and constructive feedback. We are encouraged that the reviewers found our theoretical framework to be a "rigorous, formal theory" (M2TT) and a "clear theoretical advance" (icAF), providing "clean" theorems (j8eg) that move diversity "from ad-hoc heuristics to a principled design dimension" (M2TT). We are also glad reviewers found our analysis "very clear and well explained" (JKoM) and that our proposed algorithm, HetGPS, is a "significant algorithmic contribution" (M2TT) demonstrating a "strong consistency check" (icAF) with the theory.

In response to the (very helpful!) feedback, we have made several updates to the manuscript, which has now been uploaded:

*   **Clarified assumptions:** We expanded **Appendix R** to discuss the scope and applicability of our theoretical assumptions (symmetry, monotonicity), clarifying when they apply and emphasizing that the constant-sum assumption is specific only to Theorem 3.3.
*   **Expanded experiments (Scalability):** We added new experiments with a larger number of agents (**11 agents** in 8v3 Tag and Matrix Games, Appendix J and L) to further validate the scalability of our theoretical predictions.
*   **Improved figure clarity:** We added tables summarizing the final heterogeneity gain values for Figure 4 (**Tables 10, 11**) and referenced them in the captions to improve the readability of the plots.
*   **Expanded Related Work:** We updated **Section 1.1** to include suggested literature on reward design and credit assignment, providing further context for our work.
*  **Clarified the notation:** We implemented the suggestions regarding making our notations clearer (e.g., minimizing the use of the $\bigoplus$ notation in favor of $T$ and $U$) in the revised paper.


We address specific comments below and welcome further discussion.

---

### Meta-Review · Area_Chair_rJwp · 2026-01-06

**Summary:**

The reviewers all agreed that the paper was clearly written and the theory put forth to determine when diversity might help in a cooperative MARL task was both insightful and practically demonstrated in experiments. There were some concerns that the authors agreed with such as the fact that the theory is "restricted to the reward structure and does not formally integrate the complexity of environment dynamics", but the current contribution is a strong initial foray into this new research direction that others might build upon.

**Reviewer Concerns:**

The authors clearly outlined the reviewer concerns that they addressed in the rebuttal and revision including clarified assumptions, expanded experiments on scalability, improved figure clarity, expanded related work, and clarified notation. No unaddressed concerns are outstanding.

**Reviewer Scores:**

- JKoM: This reviewers score was already high, so I do not expect they would have increased it.
- j8eg: They would have either kept their score or increased it by +1 to 7.
- M2TT: They would have either kept their score or increased it by +1 to 7.
- icAF: They probably would have increased it by +1 to 7.

---

### Decision · Program_Chairs · 2026-01-26

Accept (Poster)